# Identification of genetic factors influencing metabolic dysregulation and retinal support for MacTel, a retinal disorder

Roberto Bonelli[1,2,9], Victoria E. Jackson[1,2,9], Aravind Prasad[1,2], Jacob E. Munro [1,2], Samaneh Farashi[1,2], Tjebo F. C. Heeren[3,4], Nikolas Pontikos [3,4], Lea Scheppke[5], Martin Friedlander [5,6], MacTel Consortium*, Catherine A. Egan[3], Rando Allikmets[7,8], Brendan R. E. Ansell [1,2] & Melanie Bahlo [1,2✉]

Macular Telangiectasia Type 2 (MacTel) is a rare degenerative retinal disease with complex genetic architecture. We performed a genome-wide association study on 1,067 MacTel patients and 3,799 controls, which identified eight novel genome-wide significant loci ($p < 5 \times 10^{-8}$), and confirmed all three previously reported loci. Using MAGMA, eQTL and transcriptome-wide association analysis, we prioritised 48 genes implicated in serine-glycine biosynthesis, metabolite transport, and retinal vasculature and thickness. Mendelian randomization indicated a likely causative role of serine (FDR $= 3.9 \times 10^{-47}$) and glycine depletion (FDR $= 0.006$) as well as alanine abundance (FDR $= 0.009$). Polygenic risk scoring achieved an accuracy of 0.74 and was associated in UKBiobank with retinal damage ($p = 0.009$). This represents the largest genetic study on MacTel to date and further highlights genetically-induced systemic and tissue-specific metabolic dysregulation in MacTel patients, which impinges on retinal health.

[1] Population Health and Immunity Division, Walter and Eliza Hall Institute of Medical Research, Parkville, VIC, Australia. [2] Department of Medical Biology, University of Melbourne, Parkville, VIC, Australia. [3] Moorfields Eye Hospital NHS Foundation Trust, London, UK. [4] University College London Institute of Ophthalmology, London, UK. [5] The Lowy Medical Research Institute, La Jolla, CA, USA. [6] Department of Molecular Medicine, The Scripps Research Institute, La Jolla, CA, USA. [7] Department of Ophthalmology, Columbia University, New York, NY, USA. [8] Department of Pathology and Cell Biology, Columbia University, New York, NY, USA. [9]These authors contributed equally: Roberto Bonelli, Victoria E. Jackson. *A list of authors and their affiliations appears at the end of the paper. ✉email: bahlo@wehi.edu.au

Macular telangiectasia type II ('MacTel') is a relatively rare (~1/1000) degenerative disease of the retina[1], with a mean age at diagnosis of 60 years, and known comorbidity with type II diabetes (T2D)[2]. The clinical features of MacTel include macular photoreceptor atrophy, retinal greying, vascular leakage and in some cases, appearance of blunted/angular blood vessels in the normally avascular fovea. MacTel patients may initially experience blurred or fragmented central vision, which may slowly progress to more significant loss of central and/or paracentral vision.

To establish the genetic architecture of MacTel, we previously undertook the first genome-wide association study (GWAS) of 476 patients and 1733 controls[3], and found three GW significant loci ($P < 5 \times 10^{-8}$). The first locus, proximal to transmembrane protein 161B (*TMEM161B*; 5q14.3), was implicated in modifying retinal vascular calibre, consistent with the abnormal retinal vascular morphology observed in MacTel[1]. The other two loci were within the phosphoglycerate dehydrogenase (*PHGDH*; locus 1p12) and carbamoylphosphate synthase 1 (*CPS1*; 2q34) genes. These loci, together with two loci attaining suggestive significance ($P < 5 \times 10^{-6}$) within phosphoserine phosphatase (*PSPH*; 7p11.2) and aldehyde dehydrogenase 1L1 (*ALDH1L1*; 3q21.3), suggested systemic perturbation of glycine and serine metabolism in MacTel patients, which was supported by significantly depleted concentrations of both metabolites in patient serum.

To better characterise the systemic and retina-specific phenotypes influenced by MacTel GWAS loci, we recently leveraged GWAS summary statistics to predict genetically derived retinal vascular calibre[4,5], T2D risk[6], as well as serum concentrations for 174 metabolites in MacTel patients compared to controls. Mendelian randomisation analysis revealed that genetically influenced serine and glycine deficiency, and excessive alanine in patient sera, are likely causal for MacTel[7]. Further, a GWAS analysis performed by conditioning on genetically predicted T2D risk, glycine and serine concentrations, identified that the genetic contributions from the vasculature-associated locus 5q14.3 was independent of these metabolic traits. Furthermore, two novel loci were detected at 3p24.1 and 19p13.2. The 3p24.1 locus lies between the retinal bicarbonate symporter gene *SLC4A7* and a neuronal transcriptional factor *EOMES*, while locus 19p13.2 is upstream of *CERS4*, which synthesises the ceramide precursors of sphingolipids.

The detrimental effect of serine insufficiency on lipid metabolism, and sphingolipid biosynthesis in particular, in MacTel, has since been demonstrated in patients with a rare neuropathy: hereditary sensory neuropathy type 1 (HSN1; MIM162400). HSN1 is caused by mutations in the *SPTLC1* and *SPTLC2* genes, encoding for the subunits of serine-palmitoyltransferase (SPTLC), which acts upstream of the sphingolipid pathway. These mutations cause excessive deoxysphingolipid synthesis. Intriguingly, HSN1 patients develop peripheral neuropathy and clinical MacTel in many, but not all, cases[8].

Heritability estimates from the original MacTel GWAS suggested that more genetic loci remained to be discovered, which could be achieved with increased sample sizes. In this study we performed a GWAS on MacTel with more than double the cases of the original 2017 study. We aimed to: (1) confirm the originally reported genetic loci, and assess the strength of association with previously suggestive-significant single-nucleotide polymorphisms (SNPs); (2) identify new disease loci; (3) validate the causal effect of metabolite disruption on MacTel disease and assess genetic correlations with other traits/diseases; (4) prioritise likely causal genes by investigating functional consequences of MacTel SNPs on transcription across relevant tissues and cell types; and (5) assess the predictive value and potential clinical utility of polygenic risk scores (PRS). A diagram of the study is presented in Fig. 1.

## Results

After quality control and imputation, genotype data for 7,289,516 SNPs across 1067 MacTel patients and 3799 controls were included in genome-wide (GW) association testing (Supplementary Data 1). The sample was predominantly of European ancestry (3745 controls, 98%; 931 cases, 87%), with a higher proportion of females (78% in controls; 60% in cases; Supplementary Data 2). Power calculations indicated that this study design had in excess of 80% power to detect GW significant associations for SNPs with MacTel population frequencies ≥0.05 and odds ratios greater than 2.0 (Fig. S2).

We performed two separate GWAS: one for the entire cohort and one restricted to individuals of European ancestry. Eleven independent GW significant disease associations in ten regions were identified in the full-cohort analysis (Fig. 2), of which seven results in six regions were preserved in the European ancestry GWAS (Table 1). There was high concordance in effect estimates for both analyses (Fig. S3), and minimal inflation of test statistics due to population stratification ($\lambda = 1.011$ in the full-cohort analysis; $\lambda = 1.013$ in Europeans only, see Fig. S4 for QQ plots and Supplementary Note 1 for LD score regression (LDSC) results). Given the overall agreement in results from these analyses, we hereafter focus on results from the more powerful full-cohort analysis. After restricting to unrelated individuals (787 cases and 2853 controls) and assuming a population prevalence for MacTel of 0.45%, we estimated the narrow-sense heritability ($h^2$) for MacTel to be 0.647 (s.e. = 0.048). Bayesian fine mapping[9] was undertaken to identify 95% credible sets of causal SNPs at each GW-significant locus. The number of SNPs in each credible set ranged from 1 to 57. The credible SNP sets are listed in Supplementary Data 3, and LocusZoom plots for all loci are provided in Fig. S5.

**Associations at previously reported MacTel risk loci and a new rare PHGDH variant.** The three GW-significant MacTel loci identified in our original MacTel GWAS[3] again reached GW significance. The most statistically significant SNPs at each locus were rs146953046 in *PHGDH* (1p12, $P = 7.90 \times 10^{-22}$); rs1047891 in *CPS1* (2q34, $P = 8.6 \times 10^{-21}$) and rs17421627 near *TMEM161B* (5q14.3, $P = 4.7 \times 10^{-17}$). Whereas the SNPs at 2q34 and 5q14.3 were in linkage disequilibrium (LD) with the previously identified tagging SNPs ($r^2 = 0.91$ with rs715, and $r^2 = 0.67$ with rs73171800, respectively), the most significant SNP in the 1p12 region (rs146953046, MAF in cases/controls: 0.059/0.016) was in low LD with, and independent from the previously identified SNP rs477992 ($R^2 = 9 \times 10^{-4}$, MAF = 0.342). Conditional analysis at this locus revealed a second signal, whose top SNP rs532303 ($P = 2.3 \times 10^{-20}$, MAF in cases/controls: 0.424/0.315) was in strong LD with the previously identified SNP rs477992 ($R^2 = 0.99$). Both signals in *PHGDH* were present in both the European-only and full-cohort analyses. The distributions of the two *PHGDH* SNPs in cases and controls are given in Supplementary Data 4.

**Novel GW significant loci.** Three loci previously implicated in MacTel with suggestive association ($P < 1 \times 10^{-5}$), reached GW significance in the present analysis. Specifically, the most significant SNPs were located in the genes *PSPH* (rs6955423 in locus 7p11.2; $P = 5.9 \times 10^{-09}$), *TTC39B* (rs677622, 9p22.3; $P = 3.2 \times 10^{-09}$), and *REEP3* (rs10995566, 10q21.3; $P = 4.8 \times 10^{-08}$). A further two loci tagged by rs9820465 between the genes *SLC4A7* and *EOMES* (3p24.1, $P = 5.6 \times 10^{-09}$), and rs139412173 near *CERS4* (19p13.2, $P = 2.9 \times 10^{-08}$), were previously implicated in a GWAS of MacTel conditioned on genetically predicted glycine and serine[7]. We identified two novel GW-significant loci tagged by rs2160387 in

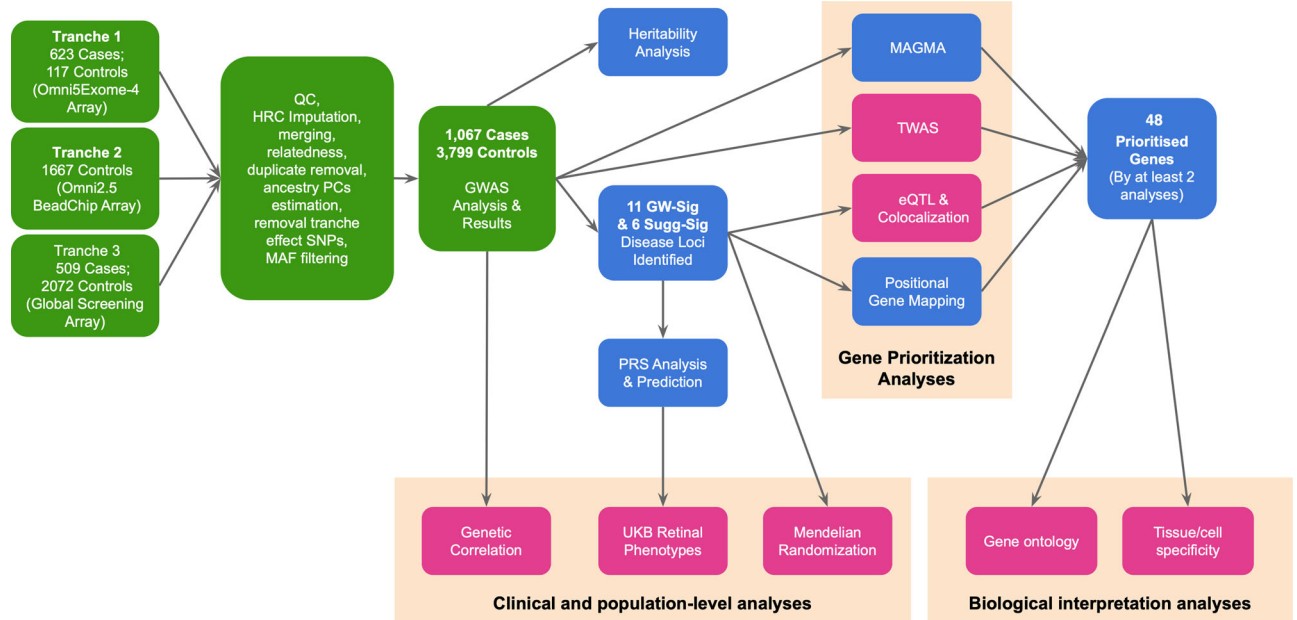

**Fig. 1 Flowchart of the analyses performed in this study.** Each box represents either a set of data or an analysis performed in this study. Green boxes highlight GWAS analyses, while the rest are post-GWAS investigations. The analyses presented in pink boxes employ both data from the current GWAS as well as publicly available data. Orange clusters represent analyses performed to achieve an overall task. GW-Sig genome-wide significant, Sugg-Sig suggestive significant, TWAS transcriptome-wide association analysis, eQTL expression quantitative trait loci, PRS Polygenic Risk Score, UKB UK Biobank. Details for the 11 GW significant and 6 suggestive-significant loci are provided in Tables 1 and 2.

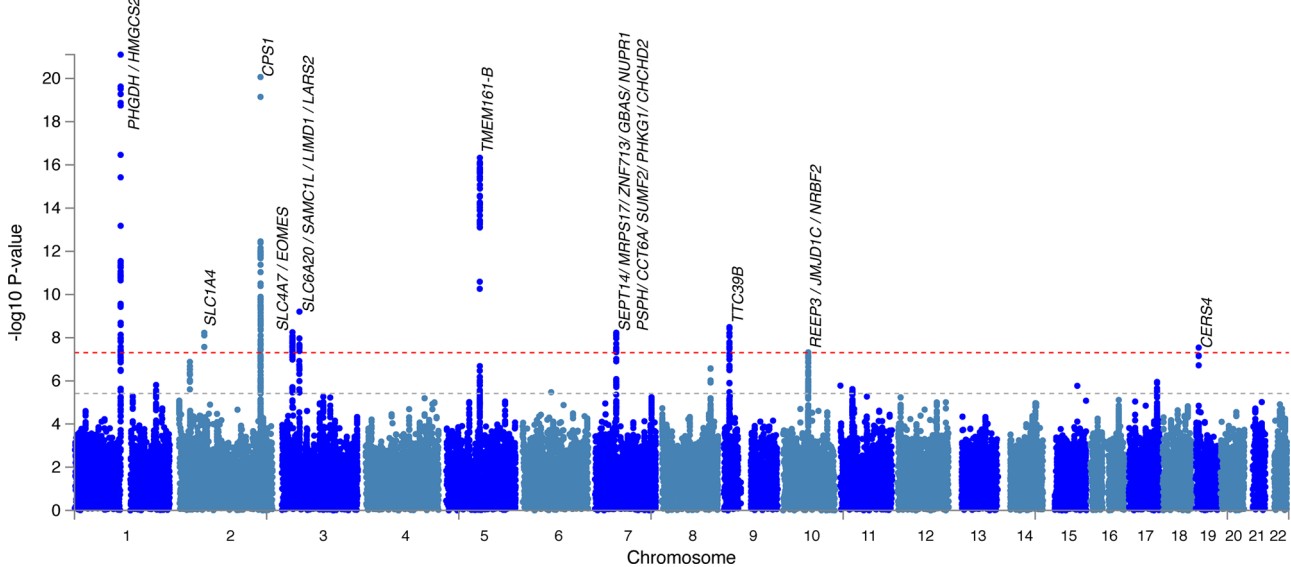

**Fig. 2 Manhattan plot displaying loci associated with macular telangiectasia type II.** Each point denotes a single-nucleotide polymorphism (SNP) located on a particular chromosome (x-axis). The significance level is presented in the y-axis. The dotted grey line indicates the threshold for genome-wide significance $5 \times 10^{-8}$ while the dotted orange line indicates the threshold for suggestive genome-wide significance $5 \times 10^{-6}$. Genes located proximal to or within genome-wide significant loci are displayed above each locus.

locus 2p14 ($P = 5.9 \times 10^{-09}$), located within the alanine/glycine/serine transporter gene *SLC1A4*, and rs17279437 in locus 3p21.31 ($P = 6.2 \times 10^{-10}$) within the glycine transporter gene *SLC6A20*. In addition to the GW-significant association in locus 19p13.2 upstream of *CERS4*, the analysis restricted to European subjects identified a second independent suggestively significant SNP (rs36259) in exon 17 of *CERS4* ($P = 1.4 \times 10^{-06}$, MAF in cases/controls = 0.294/0.246); however, this did not meet the suggestive significance threshold in the full-cohort analysis ($P = 1.4 \times 10^{-05}$, MAF in cases/controls = 0.286/0.246). A further five novel loci that attained suggestive significance are detailed in Table 2.

**Sex interaction analyses.** We undertook sex-specific association analyses for the 11 GW significant SNPs, and formally tested for an interaction between the male and female effects (Supplementary Data 5 and Fig. S6). A significant sex interaction was found for rs1047891 at 2q34 ($P_{\text{interaction}} = 1.06 \times 10^{-4}$), with a significant effect seen in females, but not males (OR = 0.45, $P = 6.43 \times 10^{-23}$ in females; OR = 0.798, $P = 0.019$ in males). Nominally significant interactions were observed for the two SNPs in the 1p12 locus, again with a larger effect in females ($P_{\text{interaction}} = 0.049$ for rs146953046; $P_{\text{interaction}} = 0.034$ for rs532303).

**Table 1 Genome-wide significant associations with MacTel.**

| Rsid | Chr | BP | Effect/non-effect allele | EAF cases | EAF controls | Cytoband | Odds ratio | 95% CI | P value | P value Eur | Proximal coding genes* | Scerri et al., 2017 GWAS | Bonelli et al., conditional GWAS |
|---|---|---|---|---|---|---|---|---|---|---|---|---|---|
| rs146953046 | 1 | 120278072 | G/T | 0.059 | 0.016 | p12 | 5.47 | 3.87–7.74 | $7.90 \times 10^{-22}$ | $1.30 \times 10^{-20}$ | **PHGDH; HMGCS2** | - | - |
| rs532303 | 1 | 120265444 | G/A | 0.576 | 0.685 | p12 | 0.585 | 0.52–0.65 | $9.29 \times 10^{-18*}$ | $1.75 \times 10^{-16*}$ | **PHGDH; HMGCS2** | GW | - |
| rs2160387 | 2 | 65220910 | C/T | 0.351 | 0.441 | p14 | 0.72 | 0.65–0.81 | $5.90 \times 10^{-09}$ | $3.70 \times 10^{-09}$ | **SLC1A4** | GW | - |
| rs1047891 | 2 | 211540507 | A/C | 0.192 | 0.306 | q34 | 0.56 | 0.5–0.63 | $8.60 \times 10^{-21}$ | $3.60 \times 10^{-20}$ | **CPS1** | GW | GW |
| rs9820465 | 3 | 27706298 | C/T | 0.145 | 0.205 | p24.1 | 0.66 | 0.57–0.76 | $5.60 \times 10^{-09}$ | $1.50 \times 10^{-07}$ | SLC4A7; EOMES | S | GW |
| rs17279437 | 3 | 45814094 | A/G | 0.144 | 0.104 | p21.31 | 1.73 | 1.45–2.06 | $6.20 \times 10^{-10}$ | $9.70 \times 10^{-10}$ | **SLC6A20; SACM1L; LIMD1; LARS2** | - | - |
| rs17421627 | 5 | 87847586 | G/T | 0.132 | 0.069 | q14.3 | 2.31 | 1.9–2.81 | $4.70 \times 10^{-17}$ | $1.00 \times 10^{-16}$ | TMEM161B | GW | GW |
| rs6955423 | 7 | 56099352 | A/G | 0.651 | 0.746 | p11.2 | 0.7 | 0.62–0.79 | $5.90 \times 10^{-09}$ | $2.60 \times 10^{-07}$ | **SEPT14;MRPS17; ZNF713;GBAS;PSPH; CCT6A;SUMF2;PHKG1; CHCHD2;NUPR1L** | S | - |
| rs677622 | 9 | 15302613 | G/A | 0.821 | 0.869 | p22.3 | 0.63 | 0.54–0.73 | $3.20 \times 10^{-09}$ | $1.10 \times 10^{-09}$ | TTC39B | S | - |
| rs10995566 | 10 | 65363166 | T/C | 0.231 | 0.316 | q21.3 | 0.72 | 0.64–0.81 | $4.80 \times 10^{-08}$ | $8.80 \times 10^{-08}$ | **NRBF2;JMJD1C;REEP3** | S | - |
| rs139412173 | 19 | 8235251 | G/A | 0.024 | 0.056 | p13.2 | 0.47 | 0.36–0.61 | $2.90 \times 10^{-08}$ | $3.40 \times 10^{-07}$ | CERS4, FBN3 | S Nearby | GW |

Top tagging GW-significant SNPs for each locus and relevant candidate genes from GWAS analysis. Bold text indicates the gene is covered by the haplotype (as defined by FUMA, based on linkage disequilibrium using the 1000 Genomes Project European cohort). Non-bold text indicates genes proximal to the locus. The Bonelli et al. GWAS* was conditioned on genetically predicted T2D risk, and serum glycine and serine. Chr chromosome number, BP base pairs (Hg19 build), EAF effect allele frequency, CI confidence interval, Eur European-only sample subset, GW, genome-wide significant ($P < 5 \times 10^{-8}$), S suggestive significant ($P < 5 \times 10^{-6}$). *P values are conditional on SNP rs146953046 (non-conditional P values in all ancestry/euro analyses: $2.38 \times 10^{-20}/5.49 \times 10^{-19}$). This table is duplicated in Supplementary Data 17.

**Gene prioritisation**. Through positional mapping, a total of 48 genes were identified within 14 MacTel-associated loci described above. An additional four genes were identified as proximal genes (closest upstream and downstream) for loci 3p24.1 and 19p13.2 (Table 1). To prioritise those genes most likely to be involved in the disease mechanism and identify additional novel genes, we performed gene-wise SNP enrichment analysis with MAGMA[10]; and inferred the transcriptional effects of each locus in multiple tissues using both eQTL analysis and transcriptome-wide association analysis (TWAS)[11].

**MAGMA analysis**. Seventy-three genes were identified by MAGMA to be enriched with MacTel-associated SNPs (FDR < 0.05) (Supplementary Data 6), of these, 29 were located within disease-associated loci, and 44 were more distal. The top ten ranked genes are provided in Table 3. Of particular interest were genes related to lipid oxidation (PPARA, SUMF2, HMGCS3), low-density lipoprotein metabolism (LDLRAD3) and cholesterol excretion (SLC51A), and a second Solute Carrier Family (SLC) gene, monocarboxylic acid transporter SLC16A8, previously implicated in age-related macular degeneration. The top ranked gene was PHGDH.

**eQTL analysis**. MacTel SNPs at each locus were investigated for associations with transcription of proximal genes (eQTL) in several tissues via the FUMA portal[12]. Specifically, eQTL for brain (16 regions), arteries (3 sites), tibial nerve, liver, EBV-transformed lymphocytes, and four whole-blood datasets were chosen via the GTEx[12], eQTLGen (eqtlgen.org)[13] and BIOS consortia[14] (Supplementary Data 7). Recently published retinal eQTLs were manually curated[15] prior to inclusion. In total, we identified 99 genes whose expression was significantly altered by MacTel-associated SNP(s), in one or more tissues (Fig. 3 and Supplementary Data 8). A heatmap of mean eQTL effects per gene per tissue can be found in Fig. S7. Positive eQTL effects on the matrix protein gene MXRA7 (17q25.1) in six tissues with a mean effect size (unit increase) of 1.82 was the strongest result, followed by zinc-finger domain protein ZNF713 (7p11.2) and the mitochondrial respiration-related gene GBAS (7p11.2). Conversely, transcriptional suppression of the sphingolipid-related gene SUMF2 (7p11.2) and differentiation factor ATRAID (2p23.3) was inferred in MacTel patients several tissues. Suppression of PHGDH (1p12) transcription was inferred in the tibial nerve, brain and vasculature. The direction of the eQTL transcriptional effect deviated markedly in the retina compared to other tissues for the genes NRBF2 (10q21.3) and TMEM161B (5q14.3) (negative in retina, positive in brain), and DCDC1 (11p14.1) (vice versa).

At the four GW significant loci which overlapped with significant retina eQTLs, we additionally tested for co-localisation of the GWAS, and eQTL signals, However, no evidence for a shared underlying causal variant was found (posterior probability <75%) (Supplementary Data 9) for any of the four loci.

To further investigate possible functions of the rare, highly deleterious SNP rs146953046, we first curated 27 tissues with significant eQTL effects on PHGDH (including 12 brain regions) via the GTEx server, and found that all normalised effects of the effect (G) allele suppressed gene expression (Fig. S8a). As this SNP was not imputed in the original retinal eQTL study by Ratnapriya et al.[15], we re-imputed the genotype data from that study and identified as heterozygotes two healthy controls and seven age-related macular degeneration (AMD) patients. Significantly lower PHGDH expression was observed in these subjects compared to homozygous reference subjects when correcting for age, sex and AMD status ($P < 0.003$; Fig. S8b). Given that spliceQTL effects are also reported for rs146953046 in

**Table 2 Novel suggestive-significant MacTel loci.**

| Rsid | Chr | BP | Effect/non-effect allele | EAF cases | EAF controls | Cytoband | Odds ratio | 95% CI | P value | P value Eur |
|---|---|---|---|---|---|---|---|---|---|---|
| rs2120770 | 1 | 212509005 | C/A | 0.179 | 0.218 | q32.3 | 0.717 | 0.63–0.82 | $1.55 \times 10^{-06}$ | $9.49 \times 10^{-06}$ |
| rs1260326 | 2 | 27730940 | C/T | 0.511 | 0.589 | p23.2 | 0.742 | 0.66–0.83 | $1.32 \times 10^{-07}$ | $5.67 \times 10^{-07}$ |
| rs1973480 | 6 | 74511980 | G/A | 0.342 | 0.295 | q13 | 1.33 | 1.18–1.5 | $3.35 \times 10^{-06}$ | $3.68 \times 10^{-06}$ |
| rs2954021 | 8 | 126482077 | G/A | 0.442 | 0.519 | q24.21 | 0.75 | 0.67–0.84 | $2.65 \times 10^{-07}$ | $2.28 \times 10^{-07}$ |
| rs2984814 | 11 | 31554964 | T/G | 0.259 | 0.325 | p13 | 0.755 | 0.67–0.85 | $2.40 \times 10^{-06}$ | $1.87 \times 10^{-06}$ |
| rs11077850 | 17 | 74661436 | C/T | 0.255 | 0.18 | q25.1 | 1.38 | 1.21–1.58 | $1.12 \times 10^{-06}$ | $1.58 \times 10^{-06}$ |

Novel suggestive-significant MacTel loci. The table presents the most significant SNPs reaching suggestive significance ($P < 5 \times 10^{-6}$) at each locus. All listed genes are covered by the haplotype (as defined by FUMA, based on linkage disequilibrium using the 1000 Genomes Project Europeans). Chr chromosome number, BP base pairs, EAF effect allele frequency, Eur European-only sample subset. This table is duplicated in Supplementary Data 18.

the skin, tibial nerve and artery, and oesophageal mucosa, we compared *PHGDH* exon usage in the retina between hetero-zygotes and reference homozygous subjects using a similar statistical framework as for eQTL testing, but found no significant differences (Fig. S8c).

**Transcriptome-wide association results**. TWAS analysis identi-fied seven genes among six tissues in which SNP-regulated expression significantly affected disease risk (FDR < 0.05) (Table 4)[11]. A further ten genes achieved suggestive significance (FDR < 0.1) (Supplementary Data 10). Increased expression of *PHGDH* and *MXRA7*, and down-regulation of nuclear receptor binding factor 2 (*NRBF2*), was inferred in MacTel patient retinae. In brain tissue, we predicted increased expression of *DNAJC24* and *SLC1A1* and suppression of the intraflagellar transport protein-coding gene *IFT172*. Whereas no differentially expressed genes were predicted in the vascular system or peripheral nerves, MacTel risk loci were associated with increased expression of *TTC39B* in the liver. Notably, all functionally-associated genes identified here fall within a locus with GW- or suggestive asso-ciation with MacTel, with the exception of *SLC1A1*, at 9p24.2.

**Gene expression patterns across body tissues and retinal cell types**. We identified 153 genes associated with MacTel via at least one of genomic location, SNP enrichment (MAGMA), eQTL or TWAS results (Supplementary Data 11). To investigate MacTel-associated gene expression throughout the body and in ocular tissues and cell types, we prioritised 45 genes supported by at least two of the above sources of evidence, and added the genes *EOMES*, *SLC4A7*, *FBN3* and *CERS4*, which were in proximity to two GW-significant loci located in gene deserts, but were not prioritised by post-GWAS analyses. We also included six solute carrier (SLC) genes for analysis (making nine in total) due to the tissue-specific expression tendency of this gene family, which could highlight disease contributions from non-retinal body tissues.

Comparison of ranked gene expression across the neural retina, brain, peripheral nerves, liver and whole blood confirmed that *SLC4A7* was the most highly expressed SLC in the neural retina whereas SLCs *6A20*, *16A8* and *51A* were much lower. *MXRA7* and *JMJD6* were also highly expressed in retina relative to other tissues. By contrast, *GCKR*, *CPS1*, *HMGCS2* and SLCs *1A1*, *5A6* and *51A* exhibited strongest expression in the liver (Figure S9). When gene expression was compared across temporal, macular and nasal regions of both the neural retina and retinal pigment epithelium (RPE)/choroid from four adult donors[16], 19 genes were relatively enriched in the neural retina, including *PAX6*, *TMEM161B*, *EOMES*, *MXRA7* and SLCs *4A7*, *1A4*, *7A1* and *4A1AP* (Supplementary Data 12 and Fig. S10). Among 20 genes showing the opposite trend were *REEP3*, *PHGDH*, *HMGCS2*, *CPS1* and SLCs *6A20*, *16A8*, *5A6* and *51A*. Tests for area specificity within tissues where genes were most expressed revealed that in the retina, *EOMES* was more highly expressed in the macula than either the nasal or temporal sides, and SLCs *1A1*, *7A1*, and *4A1AP* were more highly expressed in the macula than the temporal side.

Using five publicly available scRNA-seq datasets from three independent studies[17–19], we tested for retinal cell specificity of the prioritised genes. Expression of the genes *PHGDH*, *TMEM206* and *PSPH* was significantly higher in Müller glia than other cell types, whereas *TMEM161B*, *SLC4A7*, *SLC6A20* and *JMJD1C* were among nine genes more highly expressed in rod photoreceptors. Conversely, *CPS1*, *SLC1A1* and *PAX6* were enriched in amacrine cells, and *EOMES* was most abundant in retinal ganglion cells (Supplementary Data 13).

**Table 3 Top ten genes enriched with SNPs signals (MAGMA).**

| Symbol | Chr | Start | Stop | ZStat | P value | FDR |
|---|---|---|---|---|---|---|
| *PHGDH* | 1 | 120202421 | 120286838 | 9.07 | $5.76 \times 10^{-20}$ | $1.06 \times 10^{-15}$ |
| *HMGCS2* | 1 | 120290619 | 120311528 | 6.17 | $3.39 \times 10^{-10}$ | $3.13 \times 10^{-06}$ |
| *REEP3* | 10 | 65281123 | 65384883 | 5.97 | $1.20 \times 10^{-09}$ | $7.42 \times 10^{-06}$ |
| *SLC6A20* | 3 | 45796942 | 45838027 | 5.74 | $4.82 \times 10^{-09}$ | $2.23 \times 10^{-05}$ |
| *TTC39B* | 9 | 15163620 | 15307358 | 5.28 | $6.58 \times 10^{-08}$ | $2.43 \times 10^{-04}$ |
| *GCKR* | 2 | 27719709 | 27746554 | 4.96 | $3.56 \times 10^{-07}$ | $1.10 \times 10^{-03}$ |
| *DCDC1* | 11 | 30851916 | 31391357 | 4.79 | $8.54 \times 10^{-07}$ | $2.25 \times 10^{-03}$ |
| *ZAP70*[a] | 2 | 98330023 | 98356325 | 4.76 | $9.73 \times 10^{-07}$ | $2.25 \times 10^{-03}$ |
| *MXRA7* | 17 | 74668633 | 74707098 | 4.73 | $1.10 \times 10^{-06}$ | $2.26 \times 10^{-03}$ |
| *PSPH* | 7 | 56078744 | 56119297 | 4.71 | $1.23 \times 10^{-06}$ | $2.28 \times 10^{-03}$ |

Top ten genes enriched with SNPs signals (MAGMA). [a]Lies outside the MacTel-associated loci.

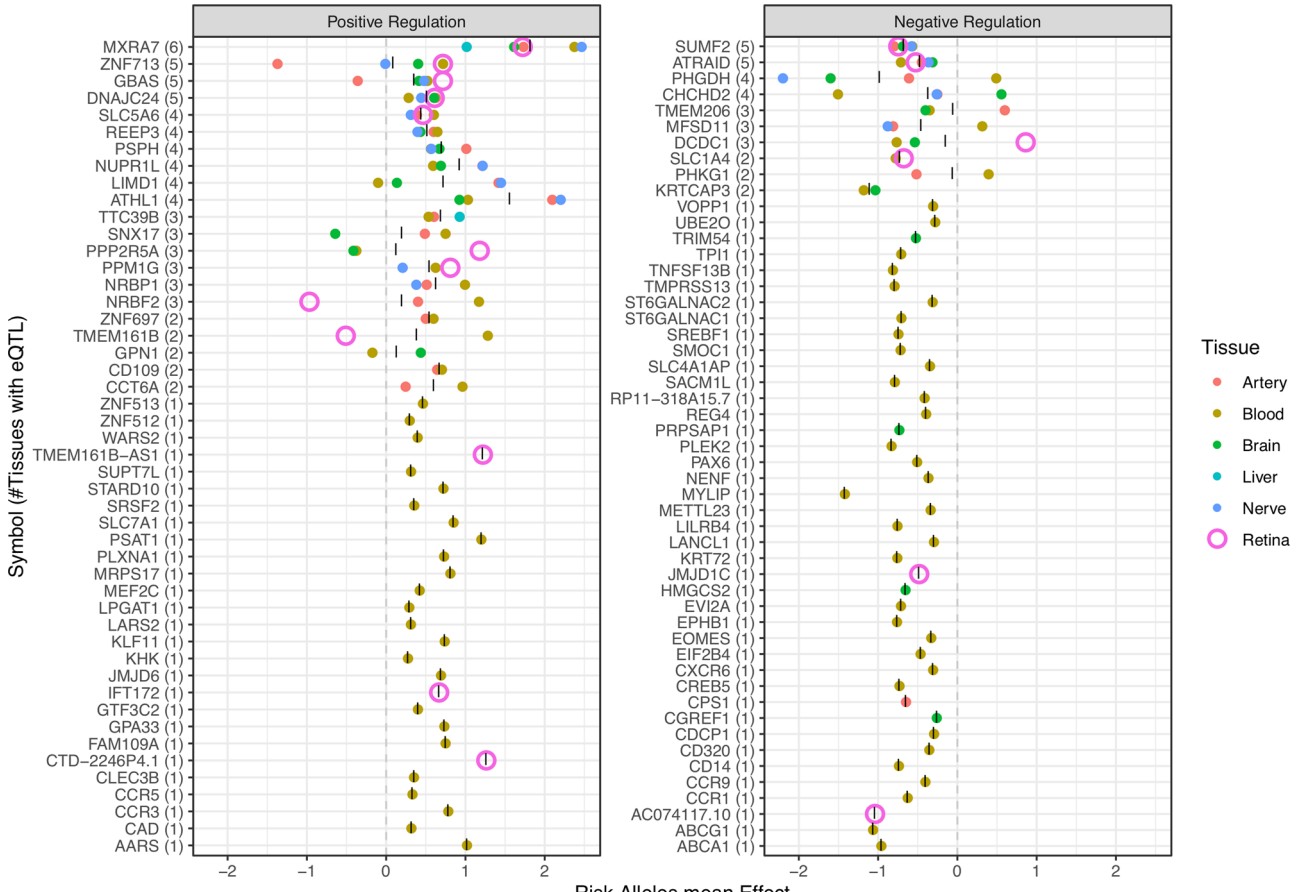

**Fig. 3 Expression quantitative trait locus transcriptional effects for MacTel risk alleles (SNPs) across six tissue types.** The averaged effects of all proximal SNPs on gene expression across tissue subtypes are represented as coloured points. As the primary disease site, retina is represented as a larger dot for ease of identification. The mean eQTL effect by tissue type is represented on *x*-axis while gene symbols are represented on the *y*-axis. Numbers in brackets denote the number of tissues in which significant eQTL were detected. Genes are ordered by the sum of tissues in which altered expression is inferred in MacTel. Genes are divided in either positive or negative global regulation defined as the average '|' of eQTL effect across the six tissue groups. *Effect on these genes was detected via trans-eQTL analysis in eQTLGen database.

**Genetic correlation between MacTel and other traits**. LD score regression between MacTel and 650 selected phenotypes revealed 99 nominally significant associations, of which use of medications for 'pain relief, constipation and heartburn' was the most significant ($P < 10 \times 10^{-5}$; Supplementary Data 14). Related to this, pain in the neck, face, limbs and joints, and neuropathy of the upper limbs were also correlated with genetic risk factors for MacTel. In addition, type II diabetes-related phenotypes,

including insulin resistance, anti-cholesterol medication, increased body fat and diabetic eye disease, were positively correlated with MacTel. Whereas triglyceride and very-low-density lipoprotein (VLDL) serum concentrations were positively associated, high-density lipoprotein (HDL) concentrations were negatively correlated with MacTel genetic risk factors. Of note, retinal detachment and breast cancer were also negatively correlated with this disease (Supplementary Data 14).

**Table 4 TWAS results in tissues relevant for MacTel.**

| Symbol | Chr | Start | End | Z | P | FDR | Tissue[a] |
|--------|-----|-------|-----|---|---|-----|--------|
| TTC39B | 9 | 15163622 | 15307360 | 5.27 | $1.39 \times 10^{-07}$ | $3.92 \times 10^{-04}$ | Liver |
| MXRA7 | 17 | 76672551 | 76711016 | 5.08 | $3.87 \times 10^{-07}$ | $3.98 \times 10^{-03}$ | Retina |
| NRBF2 | 10 | 63133247 | 63155031 | −5.01 | $5.50 \times 10^{-07}$ | $3.98 \times 10^{-03}$ | Retina |
| DNAJC24 | 11 | 31369840 | 31431849 | 4.54 | $5.56 \times 10^{-06}$ | $2.97 \times 10^{-02}$ | Brain |
| SLC1A1 | 9 | 4490444 | 4587469 | 4.27 | $1.94 \times 10^{-05}$ | $3.94 \times 10^{-02}$ | Brain |
| IFT172 | 2 | 27444371 | 27489789 | −4.24 | $2.21 \times 10^{-05}$ | $3.94 \times 10^{-02}$ | Brain |
| PHGDH | 1 | 119648411 | 119744226 | 4.45 | $8.60 \times 10^{-06}$ | $4.15 \times 10^{-02}$ | Retina |

TWAS results in tissues relevant for MacTel. *Z*, normalised effect of gene expression on disease risk (positive increases disease risk); *FDR*, Benjamini–Hochberg corrected *P* values. [a]TWAS weights for tissues accessed from GTEx (liver)[58], Rantapriya et al. (retina)[15] and Gandal et al. (brain)[59].

**Mendelian randomisation**. Using Mendelian randomisation analysis we recently demonstrated that low serine and glycine levels, as well as high alanine levels, were likely to have a causal effect on MacTel aetiology[7]. Applying Mendelian randomisation analysis to these new GWAS results we found that genetically predicted serum serine concentrations was again the strongest and most significantly associated trait among 141 predicted metabolites (Supplementary Data 15). Indeed, MacTel risk was found to nearly halve for each standard deviation unit increase in serine ($OR = 0.55$, $FDR = 3.9 \times 10^{-47}$). We identified an additional 24 metabolites to be significantly associated with MacTel at FDR < 0.05. However, conditional analysis allowed the discrimination of independent signals, and only glycine ($OR = 0.81$, $FDR = 0.006$) and alanine ($OR = 1.16$, $FDR = 0.009$) remained significant after inclusion of serine in the regression model, highlighting the high correlation between the genetic determinants of metabolites. Consistent with the previously published results, we found the T2D PRS to be robustly associated with disease risk ($OR = 1.21$, $P = 7.6 \times 10^{-07}$), and minimal confounding from pleiotropic effects (Supplementary Note 1). We additionally found a positive and significant association between retinal arteriolar calibre PRS ($b = 0.18$, $P = 1.4 \times 10^{-6}$), retinal vascular calibre PRS ($b = 0.17$, $P = 1.1 \times 10^{-5}$) and macular thickness PRS ($b = 0.08$, $P = 0.042$) with MacTel. However, in all three traits these associations were entirely driven by the shared signal in locus 5q14.3 (Fig. S11). In fact, when the 5q14.3 tagging SNP rs17421627 was included in the model as covariate the first two of these associations became non-significant while the third showed negative association (Supplementary Note 2). This negative association was once again entirely driven by the shared signal in locus 3p21.31 between the two traits (Supplementary Note 2).

**MacTel prediction and PRS**. We tested the predictive power of two separate PRS for MacTel, constructed using increasing numbers of SNPs (Fig. 4a; Methods). A conservative PRS that safeguards against overfitting, comprising only 11 SNPs tagging GW significant loci, achieved an area under the receiver operating curve (AUC) of 0.74 (CI 0.72–0.76). When trained on a set of European ancestry individuals, this classifier performed similarly in predicting the disease state of both European and non-European subjects, albeit with a modest decrease in accuracy for the latter group ($AUC = 0.78$ vs $AUC = 0.76$) (Fig. 4b). The distribution of scaled MacTel PRS was markedly higher for cases than controls (Fig. 4c). Furthermore, the odds ratio for disease risk between different PRS deciles highlighted subjects in the 10th decile of the distribution, who had an OR almost 10 times higher than those in the 5th decile. In contrast, the odds ratio for subjects in the first PRS decile was 0.25 times that of the fifth decile (Fig. 4d).

**Investigation of MacTel PRS in UK Biobank data**. Given the good prediction power of the MacTel PRS, we applied the 11 SNPs PRS to subjects from the UK Biobank database. The MacTel PRS was associated with a significantly higher probability of self-reported diagnosis with 'Other serious eye conditions' ($b = 0.048$, $P = 2.1 \times 10^{-7}$) and 'Macular degeneration' ($b = 0.029$, $P = 0.024$). Interestingly we found a negative association with self-reported 'Diabetes-related eye disorders' ($b = -0.034$, $P = 0.018$). The MacTel PRS was positively, but non-significantly, associated with reporting any eye disorder ($b = 0.007$, $P = 0.153$) which included those mentioned above, as well as 'Cataract', 'Glaucoma' and 'Injury or trauma resulting in loss of vision'.

We compared the results of manual, blinded, MacTel grading performed by expert clinicians from retinal scans on 200 randomised patients (100 in the top PRS decile and 100 in the bottom PRS decile) who self-reported absence of any eye conditions. Of these 14 retinal scans were discarded due to low scan quality which could not be scored. We did not identify any definite MacTel patients among these subjects. However, using logistic regression modelling we found the MacTel PRS decile to be positively associated with retinal phenotypes typical of MacTel disease ($b = 0.54$, $P = 0.003$) as well as retinal phenotypes not directly related to MacTel ($b = 0.17$, $P = 0.036$) compared to healthy eyes.

## Discussion

MacTel was first clinically defined in 1977; however, the associated genetic risk factors remained elusive for decades. In 2017 the first genetic association signals for MacTel were identified, highlighting glycine and serine metabolism as a dysregulated pathway in MacTel patients. These preliminary genetic clues were discovered with a small scale GWAS of 476 cases and 1733 controls. Subsequent work highlighted that further GWAS, with increased sample size, would likely increase the number of associated loci[7]. Here we describe results from a second GWAS for MacTel disease with a cohort of 1067 MacTel patients and 3799 controls for this relatively rare eye disease (approximately 250 times less common than AMD).

Given the rarity of this disease, the current study includes data from our initial GWAS[3] and as such cannot be considered a full independent replication. Despite this limitation, the increased sample size combined with improved genotype imputation results, and newly applied advanced post-GWAS techniques and statistical correction for related individuals, allowed us to generate substantial new insight into the genetic architecture of MacTel.

We made use of publicly available data sources such as GTEx, metabolomics data, retinal GWAS studies and the UK Biobank to extend and contextualise our findings, providing further clues as to biological mechanisms that are perturbed in MacTel. These results confirm previous findings, revealing loci and candidate genes that we expected to observe with an increased sample size based on the current understanding of MacTel, and additionally identified several unanticipated novel associations. We found 11

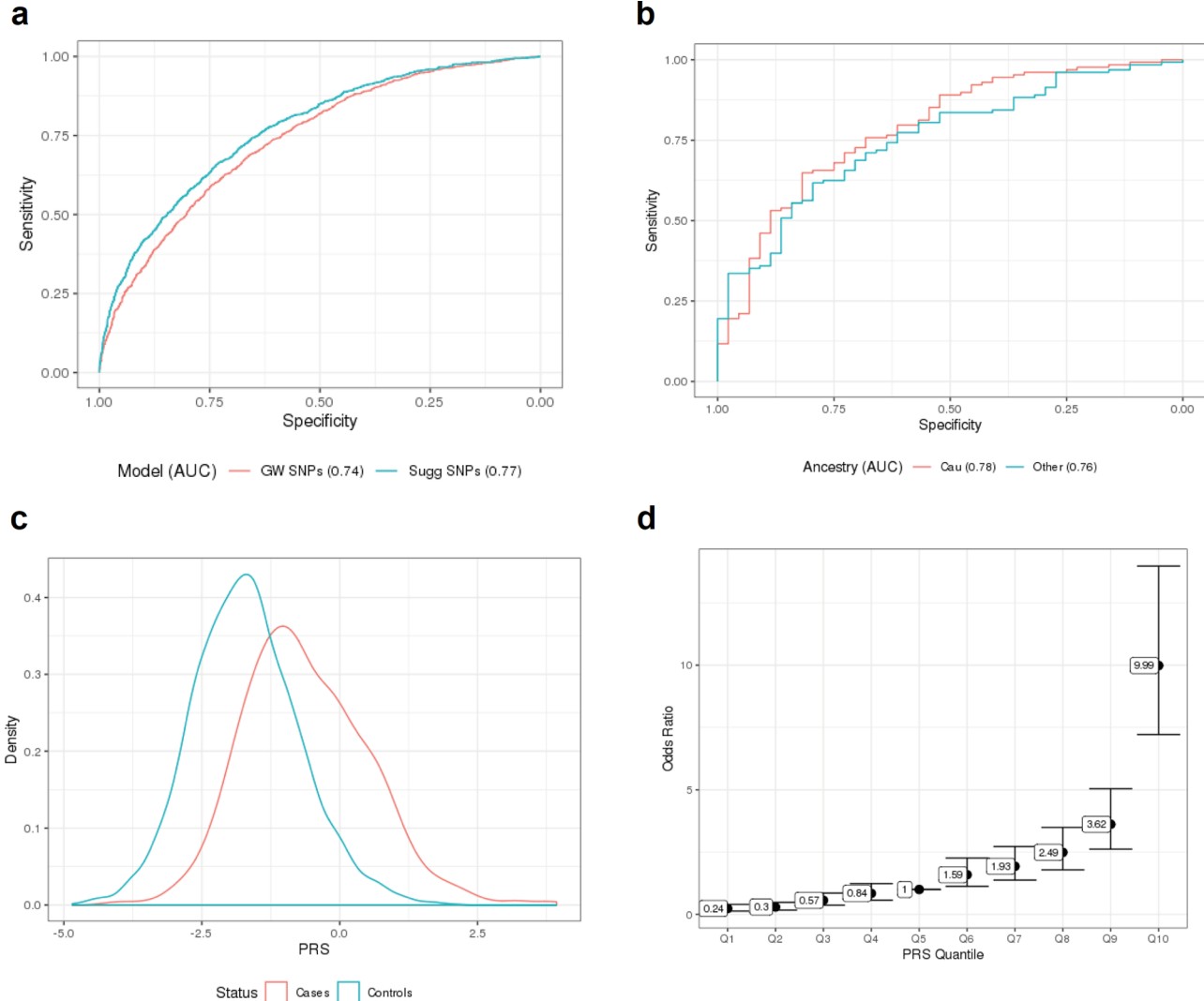

**Fig. 4 Prediction of MacTel status using polygenic risk score (PRS). a** ROC curves of the two tested PRSs. Area under the ROC curve (AUC) is reported in the figure legend. **b** Comparison of ROC curves for prediction accuracy among European and non-European. Area under the ROC curve is reported in the figure legend. **c** Density plot comparing PRS distribution of between cases and controls. **d** Odds ratio for magnitude MacTel Risk comparing deciles to the median decile.

GW-significant SNPs at ten loci, which collectively explained 0.647% of MacTel heritability, with odds ratios for significant SNPs ranging from 1.39 to 5.47. A PRS comprising all GW-significant SNPs achieved an AUC of 0.74 (CI 0.720–0.755), with patients in the top PRS decile harbouring a tenfold greater risk of MacTel compared to the fifth (middle) decile. As MacTel manifests relatively late in life and is difficult to diagnose, the genetic risk predictor provided here may be useful for both clinical diagnosis and interpretation of clinical trial results[20]. Genetic prediction of MacTel in the independent UK Biobank data demonstrated that the MacTel PRS could be of clinical use to detect early retinal damage. The model could likely be improved through integrating direct metabolic measurements, which are likely to more comprehensively describe the metabolic dysregulation known to be present in MacTel patients but which is tempered by the greater measurement error in metabolites in comparison to genotypes.

Consistent with our previous findings, some of the strongest signals from the present work highlight the involvement of glycine and serine in MacTel disease. The most significant signals observed in our study are located in the 1p12 locus, within the *PHGDH* gene.

The enzyme phosphoglycerate dehydrogenase (PHGDH) catalyses the conversion of 3-phosphoglycerate into 3-phosphohydroxypyr-uvate, which is the first step in serine biosynthesis. The strongest GW-association signal is driven by a rare risk haplotype with a frequency of 1.6% in controls and 5.9% in cases (OR = 5.47), which is independent of the previously identified common risk haplotype at this locus[3] (replicated here with an odds ratio of 0.585). The newly reported SNP was not detected in our previous study, for which SNPs with MAF below 5% were excluded due to the modest sample size. The rare risk haplotype, tagged by the SNP rs146953046, associates with lower *PHGDH* expression in the retina. The presence of two independent risk haplotypes associated with *PHGDH*, and support for involvement of this gene in MacTel from SNP enrichment, eQTL and TWAS results, indicates a central but complex role in disease aetiology. *PHGDH* is strongly expressed in the liver, the major tissue responsible for production of the deoxysphingolipids associated with retinal degeneration in serine-deprived animal models of MacTel[8]. However neither of the two risk haplotypes on chromosome 1 were predicted to affect *PHGDH* abundance in the liver. Instead, averaged eQTL effects indicated lower expression of this gene in the nervous and vascular systems

(Fig. 3). A more sensitive estimate of aggregate MacTel genetic effects on *PHGDH* expression (TWAS) indicated increased expression only in the neural retina. We must however caution the interpretation of the latter result, given (a) the absence of the rare suppressive eQTL from summary statistics used to estimate TWAS effects, and more saliently, (b) the relatively lower importance of the neural retina than the RPE for serine and sphingolipid metabolism. Specifically, *PHGDH* expression is five times higher in the human RPE than in neural retina[16] (Fig. S10), and in mice both serine and sphingolipid abundance are significantly elevated in this tissue[21]. Thus it may be that neuroretinal dysfunction is a secondary consequence of the RPE degradation. Other neuronal support cells such as Müller glia (where *PHGDH* expression is also elevated; Supplementary Data 12) may also contribute to retinal dysfunction in MacTel[22]. Further work to establish an eQTL reference for the adult human RPE, together with tissue- and cell-type-specific perturbation of *PHGDH* in experimental animals, are now required to define the key tissues driving the serine-associated axis of MacTel.

The 2q34 locus, encompassing the 3′ UTR of carbamoylphosphate synthase 1 (*CPS1*), is the strongest reported modulator of serum glycine abundance[23] and was the second-most significant MacTel-associated locus. Abundant glycine, which is converted to serine by serine hydroxymethyltransferase, may protect against MacTel[7,8]. The tagging SNP at 2q34 is a coding SNP in *CPS1* (rs1047891), with the minor allele being protective for MacTel. The resulting T1406N substitution reduces the binding affinity between CPS1 and its coactivator *N*-acetylglutamine (NAG). NAG promotes glycine flux through the urea cycle[24,25], which may contribute to MacTel by drawing down subjects' capacity to buffer serine. Indeed, urea cycle dysregulation is observed at the metabolite level in MacTel patient serum[26]. An allosteric disease mechanism affecting CPS1 enzyme activity would be consistent with the lack of eQTL effects for *CPS1* (suggesting stable enzyme abundance). However, the nominally significant TWAS effect linking greater *CPS1* transcription in the retina with MacTel suggests potentially compounding effects of greater expression and activation of *CPS1* on this disease. The relatively greater protective effect of the rs1047891 'A' allele in females, which has been previously documented[3,27], remains unexplained. This finding may nevertheless present an opportunity for the development of sex-specific clinical interventions to maximise patient benefit.

The GW significant signal at 7p11.2 spans the phosphoserine phosphatase (*PSPH*) gene. *PSPH*, along with *PHGDH* and *PSAT1* are essential genes for serine biosynthesis. While we did not identify any GW-significant signals in *PSAT1*, trans-eQTL analysis in blood revealed a possible effect of polymorphisms in *PHGDH* and *CPS1* on *PSAT1* expression, indicating co-regulated expression of genes across the serine biosynthesis pathway. The causal role of serine and glycine depletion on MacTel is also highlighted in previous investigations using Mendelian randomisation[7,23]. Replication of those findings in the present study further emphasises that genetically driven depletion of these metabolites is to likely cause MacTel.

The present study indicates involvement of sphingolipid and cholesterol metabolism in MacTel disease. Serine deficiency is implicated in retinal degeneration via the accumulation of deoxy-sphingolipids, which must be degraded via the omega oxidation pathway instead of sphingosine phosphate lyase[8]. Omega oxidation is mediated via PPAR-alpha, which can be stimulated with Fenofibrate, and protects against deoxysphingolipid-induced toxicity in retinal organoids[8]. Interestingly, our study prioritised *PPARA* via MAGMA analysis, and implicated several related genes, including *HMGCS2* (GWAS and MAGMA), *ABCA1*, *ABCG1* and *SREBF1* (trans-eQTL in blood from locus 2p23.2). Furthermore, the 10q21.3 locus, previously found to significantly affect serine and glycine levels[23], also encompasses *NRBF2*, a co-regulator of PPAR-alpha.

Two further loci (19p13.2 and 7pp11.2) overlapped with genes involved in sphingolipid metabolism (*CERS4* and *SUMF2*, respectively). These complementary lines of evidence support the centrality of perturbed lipid turnover and concomitant deoxy-sphingolipid accumulation in MacTel. The top SNP at the 9p22.3 locus is located in an intron of the Tetratricopeptide repeat protein 39B (*TTC39B*), and is a significant eQTL for this gene in the liver. TTC39B is known to regulate HDL in mice[28] and is associated with human circulating HDL concentrations via GWAS[28]. Genetic correlations between MacTel and use of cholesterol-lowering medication, increased body fat, increased HDL cholesterol levels and decreased VLDL cholesterol provide further support for the importance of perturbed lipid metabolism in MacTel.

Since disruption of amino acid and lipid metabolism appears central to MacTel aetiology, in this study we closely examined the transmembrane SLC family proteins which transport these molecules between sites of synthesis and utilisation, and which represent attractive drug targets[29]. Enrichment of disease-related SNPs in the intestinally expressed bile acid exporter *SLC51A* (implicated by MAGMA) may affect serum concentrations of cholesterol which is a major bile component, and further supports the likely perturbation in cholesterol metabolism evidenced above[30]. *SLC1A4* was implicated by several lines of evidence, and is strongly transcribed in, but not restricted in, expression to the neural retina. This gene transports serine, among other amino acids, and patients with mutations therein exhibit phenotypes similar to those in primary serine deficiencies[31]. Together with alterations in serine transport into the retina, acidification of the subretinal space may be an important driver of MacTel retinopathy. *SLC4A7* was not identified by post-GWAS analyses but sits proximal to the intergenic MacTel risk locus 3p24.1. Of the nine SLCs prioritised for investigation, *SLC4A7* shows the strongest retinal expression, in particular the rod photoreceptors, and functions to buffer pH in the subretinal space. Further work is required to determine whether this gene is under trans-regulation in the retina from the proximal MacTel risk locus. Similarly, *SLC1A1*, which is expressed moderately in the retina but strongly in the liver, transports the amino acids glutamate and aspartate. Glutamate is required for serine biosynthesis by PSAT1 (ref. [32]) and for neurotransmission. As the concentrations of glutamate and aspartate can influence extracellular pH, defective *SLC1A1* function might affect both neurotransmission and the effective transport of other nutrients between the retina and RPE.

As the predominant ocular tissue for serine metabolism and sphingolipid metabolism, the systemic and local MacTel disease processes that impinge on these metabolites may converge in the RPE. Expression of *SLC6A20* and *SLC16A8*, now implicated in MacTel genetic aetiology, is strongly biased towards the RPE rather than the retina. *SLC6A20* transports sarcosine, which is converted to glycine in a single enzymatic step involving *S*-adenosylmethionine, whose metabolic disruption is also reported in MacTel patients[26]. *SLC16A8* enables transport of lactic acid out of photoreceptor cells into the RPE, and is associated with AMD[33]. Ablation of this gene in mice leads to acidification of the outer retina and reduced light sensitivity[34]. The diverse functions and tissue distribution of the SLCs detailed here support the notion that MacTel comprises both systemic and local disease processes which manifest in the retina.

Finally, we observe that MacTel retinae are likely to be affected by genetic mechanisms other than metabolic dysfunction. The strongest signal in the preliminary GWAS for MacTel was an association peak at 5q14.3 located between the genes *TMEM161B* and *LINC00461* (OR = 2.34, *P* value = 1.08e−15). This association is further strengthened in the current study (*P* value = 4.7e−17, OR = 2.31, all ancestries). Despite its intergenic location, the 5q14.3 locus is transcriptionally active in human CNS cell

cultures, and was previously identified to affect retinal vascular calibre[4,5]. The top SNP at this locus (rs17421627) is highly conserved and modulates the expression of microRNA miR-9-5 in zebrafish (orthologous to human MIR-9-2)[35]. Genomic ablation of 770 bp covering this SNP reduces the number and branching of blood vessels in the zebrafish retina, which is phenocopied by miR-9-5 ablation[35]. This functional link with retinal vasculature formation, combined with shared genetic effects on several human retinal vascular traits, strongly suggests that locus 5q14.3 underpins the structural vascular defects characteristic of MacTel. Also reported to affect retinal vascular calibre is the gene *MXRA7* (encoding a matrix-related protein)[5]; this gene was prioritised in our study by MAGMA, and produced both the largest number of eQTL signals and the strongest TWAS association in the retina. Association between retinal vasculature defects and MacTel comes as no surprise given that the disease pathology and relationship between vascular calibre and MacTel has been previously identified[1]. Our results also shared two signals with a recent GWAS on retinal thickness, reported in UK Biobank subjects by Gao and colleagues[36]. The first signal overlapped the 5q14.3 locus, and the second was identical to our newly identified signal at the 3p21.31 locus, containing the *SLC6A20* gene. MacTel patients often present with decreased macular thickness resulting from degeneration of retinal layers. However, while the signal in the 3p21.31 locus indeed suggests that the MacTel risk allele may reduce macular thickness in UK Biobank subjects, the risk allele at 5q14.3 is in fact associated with increased thickness, and is concordant with our previous results[7]. Lastly, a recently published meta-analysis[37] identified a SNP in the 5q14.3 locus that also affects AMD risk. This SNP (rs17421410) is in strong LD with the top SNP in that locus ($r^2 = 0.725$) and, concordant with our study, also increases disease risk for AMD (OR = 1.16).

Gantner et al.[8] demonstrated that serine dietary depletion is sufficient to cause a MacTel-like ocular phenotype in a mouse model. This finding coupled with the present study results suggests that sufficient systemic serine abundance is required to maintain retinal health. While dietary supplementation of serine to MacTel patients may be a reasonable therapeutic approach, recent work suggests that local delivery or frequent systemic dosing will be required to maintain adequate L-serine levels[38]. Given the complexities of the genetic basis of MacTel and the diversity of the patient population, we would caution against serine supplementation for MacTel patients at this time, but note that its potential is currently being investigated. Although MacTel is a relatively rare retinal disorder, a better understanding of its genetic aetiology is also informing us of crucial tissue-specific metabolic and structural characteristics of the retina and RPE in general. As this GWAS study also demonstrates, a better understanding of the biological mechanisms involved in MacTel will also assist with our understanding of other, related neurovascular-glial degenerative diseases of the retina.

## Methods

**Participants**. A total of 1132 Mactel cases were recruited from 23 clinical centres, from seven countries. A complete list of participating institutes and approving ethics committees is provided in Supplementary Note 1. Participants were given a standardised ophthalmic examination, including best-corrected visual acuity, fundus photography, fluorescein angiography, optical coherence tomography and blue light reflectance. Images were adjudicated at the Reading Center at Moorfields Eye Hospital, London. Diagnoses were made in accordance with the criteria described by Clemons et al.[2] on the basis of Gass and Blodi[39]. See the Supplementary Methods for additional details on patient diagnosis. Retinal images were assessed for loss of transparency in the perifoveal region, dilated and telangiectatic blood vessels, especially in the temporal retina, and crystalline deposits.

In all, 297 unaffected family members and spouses of cases were also recruited via the MacTel Consortium as controls. There were a further 22 samples recruited via the MacTel consortium, and genotyped for this study, but whose MacTel status was subsequently reclassified as unconfirmed. The remainder of controls came from the Aged-Related Eye Diseases Study (AREDS, $n = 1657$), and an Australian Study of Twinning known as the QIMR Twinning Genetics Study ($n = 1892$, unpublished). The QIMR Twinning Genetics study was approved by the Human Research Ethics Committee of the QIMR Berghofer Medical Research Institute. The AREDS study includes individuals aged 55–80 years who 'are all Caucasian, do not have age-related macular degeneration (AMD) and were further screened to also exclude individuals with cataracts, retinitis pigmentosa, colour blindness, other congenital eye problems, LASIK, artificial lenses, and other eye surgery.' These controls were accessed via dbGaP (phs000429.v1.p1) A subset of 476 MacTel Cases, 76 controls from the MacTel Consortium, and 1657 controls from AREDs were used in our previous discovery GWAS[3].

**Genotyping**. MacTel Consortium samples were genotyped on either the Illumina Human Omni5-Exome-4 Array (623 cases; 117 controls; 12 unconfirmed phenotype) or the Illumina Global Screening Array (509 cases; 180 controls; 10 unconfirmed phenotype). AREDs controls were genotyped on the Illumina Omni2.5 BeadChip array, and the Australian Twinning controls on the Illumina Global Screening Array. Full details of samples and genotyping platforms are provided in the supplement (Supplementary Data 1 and 2). All genomic loci are reported relative to the Hg19 genome build.

**Quality control and imputation of genotyping data**. Quality control and imputation were carried out in three tranches, determined by genotyping array: Tranche 1 representing the MacTel consortium samples genotyped on Omni5-exome; Tranche 2 the AREDs controls genotyped on Omni2.5; and Tranche 3 the MacTel Consortium and Twinning samples genotyped on the Global Screening Array.

Within each tranche, initial quality control of the genotype data was undertaken using Plink (v.1.9.3) to remove SNPs with call rates below 98%, and samples with call rates below 98%, outliers in heterozygosity; samples whose sex could not be confidently genetically inferred, and those whose inferred sex did not match recorded sex. Tranches 1 and 3 contained both cases and controls, and included samples genotyped in different batches. In these two tranches additional exclusions were made of SNPs showing differential levels of missingness in cases versus controls, and SNPs showing very extreme differences in allele frequency between genotyping batches (difference in frequency between batches $P < 1 \times 10^{-10}$). Each tranche was merged with the 1000 Genomes reference dataset, and principal components (PCs) of ancestry were estimated using Plink. We identified a subset of our sample as European (931 cases and 3745 controls), whose first two PCs of ancestry were within 4 standard deviations of the mean value for the 1000 Genomes European super-population. No exclusions based on ancestry were made.

The three tranches were separately imputed to the Haplotype Reference Consortium reference panel (v.1.1), using the Michigan Imputation Server v1.0.4 pipeline[40], with phasing using Eagle[41]. The three tranches were merged following imputation.

We examined relatedness and ancestry of our samples in the merged dataset. The data were restricted to variants directly genotyped across all three tranches (230,201 SNPs), and then LD-Pruned to leave 100,804 SNPs. Using this subset of SNPs, we generated an initial genetic relatedness matrix (GRM) using the KING method[42], implemented in the SNPRelate R package[43]. Using this GRM, we then calculated PCs of ancestry, taking into account relatedness, using the PC-AiR method[44]. We then re-estimated the GRM, taking into account PCs of ancestry, using the PC-Relate method[45], in the GENESIS R package[46]. Eleven individuals were identified as duplicates, and one sample from each duplicate pair was removed, retaining the sample with the highest call rate.

Finally, we identified and removed SNPs which may have been subject to batch effects. After restricting to European samples, we tested for differences in allele frequency, for the following comparisons: tranche 1 cases versus tranche 3 cases; tranche 1 controls versus all other controls; tranche 2 controls versus all other controls; tranche 3 controls versus all other controls. SNPs which showed association ($P < 5 \times 10^{-5}$) in any tranche comparison were excluded. SNPs with imputation quality <0.9 in any imputation run were also excluded, along with any SNPs deviating from Hardy Weinberg equilibrium ($P < 5 \times 10^{-7}$).

**GW association analyses**. Association analyses were undertaken using SAIGE v.0.36.2, which allows incorporation of related individuals, which was not performed in the first MacTel GWAS[3].

Firstly, a null logistic mixed model was fitted, using a subset of 230,085 SNPs, which were directly genotyped across all samples, with sex and 8 PCs of ancestry as covariates. Eight PCs were chosen, justified from the following criteria: (1) inspection of a scree plot (Fig. S1A), and (2) through fitting the null model with a varying numbers of PCs, and selecting the best model, based on the Akaike information criterion (Fig. S1B). Using the parameters estimated under the null model, association testing was undertaken GW on the imputed data. Two GW analyses were carried out: firstly, using a subset of European ancestry individuals only; secondly, utilising all samples, regardless of ancestry ('full-cohort' analysis). Results of both analyses were filtered to include only SNPs with MAF > 0.25%.

Unless otherwise stated, SNPs with uncorrected $P$ values of association below $5 \times 10^{-8}$ were considered 'genome-wide significant' (GW significant), and those with $P < 5 \times 10^{-6}$ were considered 'suggestive significant'. Loci with suggestive-significant association with disease but supported by fewer than two SNPs were discarded.

To verify that our results were not subject to confounding by population stratification, we estimated the genomic inflation factor ($\lambda$) to assess overall inflation of the association test statistics. We additionally utilised LD score regression[47], as implemented by LDSC software version 1.1, and using LD-scores estimated from the data. After restricting the results to SNPs with MAF $\geq 1\%$, we examined the resulting LD Score regression intercept, as a measure of inflation due to confounding, rather than polygenicity of the trait.

SNP-based heritability was estimated using GREML[48,49], as implemented by GCTA[50]. Firstly, a GRM was estimated using all SNPs, and a subset of unrelated individuals (defined as pairwise kinship coefficient <0.018; 986 cases and 3227 controls). Narrow-sense heritability was estimated using GCTA-GREML, using this GRM and assuming a population prevalence for Mactel of 0.45%.

**Identification of secondary signals, sex interactions and fine mapping of causal variants**. Conditional analysis was carried out to identify secondary, independent signals. For each locus with a GW significant association, we performed conditional analyses over a 2 Mb region (±1 Mb of top SNP), conditional on the top SNP. We identified secondary signals as SNPs with $P < 5 \times 10^{-8}$, after conditioning on the original top SNP. To confirm independence where a secondary signal was found, we re-tested for association of the original top SNP, conditional on the secondary top SNP.

For the top SNPs at each identified locus, we undertook analyses in males and females separately, with adjustment for eight PCs of ancestry. We formally tested for a sex interaction using Welsch's $t$-test (Supplementary Note 1 and Supplementary Methods); $P$ values $< 4.5 \times 10^{-3}$ (Bonferroni correction for 11 SNPs) were considered significant evidence of a sex interaction.

For each identified locus, we carried out Bayesian fine mapping, to determine 95% Credible Sets of causal SNPs[9]. This method assumes a single causal SNP at each locus; for loci where a secondary, independent signal was identified, we undertook fine mapping separately for each signal, using the conditional association results.

**Functional mapping and annotation with FUMA**. For all GW significant ($P < 5 \times 10^{-8}$), and suggestive-significant loci ($P < 5 \times 10^{-6}$), gene mapping was performed using the web-tool FUMA v1.3.5e[12]. The European 1000 Genome Project reference panel was used to define LD between SNPs. A full list of all parameters used for the FUMA analysis is available in Supplementary Data 7. Firstly, we used positional mapping to identify genes containing SNPs in LD ($r^2 > 0.4$) with the most significant SNP at each locus. For intergenic loci, where neither the top SNP nor any proxy ($r^2 > 0.4$) overlapped with any genes, we manually annotated these loci with the closest genes on both sides. Secondly, gene-based analyses were carried out using MAGMA[10]. We corrected for false discovery rate the MAGMA $P$ values using an ad hoc Benjamini–Hochberg correction from the function p.adjust in R v. 3.6.1. Genes with a corrected $P$ value less than 0.05 were considered significantly enriched. Thirdly, the most significant SNPs at each locus as well as SNPs in LD ($r^2 > 0.4$) with them were mapped to significant eQTLs as defined by the internal FUMA pipeline. In order to maintain results as relevant as possible to MacTel disease only specific tissues were selected for eQTL queries. Such tissues were selected to represent those ones likely affected by the systemic dysregulations previously observed among MacTel patients. Datasets included in the eQTL analysis spans across brain, blood, liver, nerve and vasculature, specific tissues subtypes can be found among the parameters used in FUMA.

**Retinal eQTL analysis**. As retinal eQTL results were not available through FUMA, we intersected the GWAS summary results with cis-eQTL data for retinal tissue (retina eQTL)[15], available as an external dataset on the[51] GTEx portal. MacTel effect alleles were first reordered to ensure all disease effects were positive (i.e., increasing risk). Next, the effect alleles and effect estimates were inverted in the eQTL results to be consistent with the MacTel allele order, as required. At each GWAS locus (GW significant, or suggestive significant), we identified genes where the top SNP at each GWAS locus, or any SNP in LD ($r^2 > 0.4$), was a significant eQTL, based on gene-specific significance thresholds, as defined by Ratnapriya et al.[15].

For the implicated genes at each of the GW-sig locus, we additionally employed Approximate Bayes Factor colocalization analysis using the R package coloc[52]. Using the estimated effect sizes and corresponding variances from the GWAS and the eQTL analyses as input, we estimated the posterior probability that the GWAS signal and the eQTL signal were the result of a shared causal variant. A posterior probability >0.75 was taken as strong support of colocalistion of the GWAS and eQTL signal[52].

In order to analyse the rare *PHGDH* SNP rs146953046, which was not reported in the original retinal eQTL study by Ratnapriya et al.[15], the genotype data for that study were generously provided by Dr. Rinki Ratnapriya and Prof. Anand Swaroop (US National Eye Institute). Updated genotypes were imputed using the Sanger

Imputation Server version (EAGLE2 v2.0.5 + PBWT)[53,54] and the Haplotype Reference Consortium panel as a reference (r1.1). The corresponding retinal RNA-seq data (GEO accession GSE115828) was aligned to the human genome (release GRCh38.91) using the STAR alignment tool (ver. 2.6.1)[55], and transcriptional abundance was calculated using featureCounts (ver. 2.0.0)[56]. Linear regression controlling for age, sex, and AMD disease status was employed to examine the expression of *PHGDH* in individuals heterozygous for rs146953046 relative to homozygous reference individuals. To test the potential splicing effect of this SNP, differential exon usage analysis was performed on the same data by merging overlapping exons and quantifying their abundance using SubRead modules flattenGTF and featureCounts respectively[56], discarding exons with fewer than 1 count per million in at least 40 subjects, and testing differential exon abundance using the limma diffSplice module[57]. Fold-change and nominal $P$ values for exons of *PHGDH* were extracted for graphical analysis.

**Transcriptome-wide association analysis**. TWAS on gene expression was inferred using the TWAS/FUSION software[11]. Given the previously reported systemic metabolic defects in MacTel patients[8] we downloaded SNPs and relative TWAS weights on gene expression for several tissues from the TWAS/FUSION website (gusevlab.org/projects/fusion/). These tissues included liver, tibial nerve, tibial artery, and whole blood from GTEx[13,14,58] as well as retinal weights from Ratnapriya et al.[15]. For this analysis brain TWAS weights were selected from Gandal et al.[59] rather than GTEx as these were estimated using a considerably larger sample size. Only genes whose expression was predicted using at least three SNPs were considered in this analysis. Due to the large number of tests, we allowed for a false discovery rate of 5%. Genes with ad hoc corrected $P$ values (using the p.adjust function in R) less than 0.05 were thus considered significant. We observed a large number of genes in the retina TWAS analysis to be skipped by the TWAS/FUSION software due to their best predictive model only containing SNPs with zero weight. To this end, all skipped genes in the retina were re-estimated using the best linear unbiased predictor (BLUP) model as defined by the TWAS/FUSION software.

**Enrichment of prioritised genes across body tissues and retinal cell types**. We prioritised a set of genes from the GWAS and post-GWAS analysis results supported by at least two of the following criteria: (1) all genes located underneath loci with SNPs reaching GW significant threshold, (2) all genes located underneath loci with SNPs reaching suggestive-significant threshold, (3) genes whose expression was affected by the identified SNPs in at least one of the analysed tissues, (4) genes identified by the MAGMA analysis, and (4) genes identified by the TWAS/FUSION software.

We then accessed and analysed previously published human adult retina datasets from: (1) bulk RNA analysis for four adult donors for temporal, macular and nasal regions of both the neural retina, and the RPE/choroid[16]; (2) average gene expression across retinal single-cell datasets from 12 (ref. [17]) and 6 samples[18], as well as (3) average retina single-cell expression data from the fovea and peripheral areas of the retina from 5 samples[19]. Genes not expressed in any areas or cell types were excluded. Gene expression in each tissue area or cell type was log transformed to obtain an approximately normal distribution. We used linear regression modelling with a random intercept for each tissue donor to compare expression between cell types, and retinal areas of interest. $P$ values for area-specific and cell-type-specific tests were independently corrected for false discovery rate as described previously.

**Genetic correlation analysis**. The genetic correlation between MacTel and 514 UK Biobank traits, 107 metabolites[60], 8 glycaemic traits[61], 4 circulating lipid concentrations[62], and 3 neurological disorders[63] was assessed using LD score regression software[64] implemented by the the LDhub database[65] in February 2020. In addition, we included as a separate analysis, 13 ocular phenotypes[66] using LDSC standalone software[47]. We have estimated genetic correlations given the strong phenotypic correlations observed between MacTel and many of the examined traits; however, we note that genetic correlation analyses performed using small sample size GWAS summary statistics has been previously discouraged[67]. We report all nominally significant correlations ($P < 0.05$).

**Mendelian randomisation**. Mendelian randomisation was performed using PRS obtained through the weighted score method on individual genetic data as described by Burgess et al.[68] (Supplementary Methods). Instrument SNPs and relative weights were derived from different studies according to the phenotype of interest. Metabolic instruments were obtained from Lotta et al.[23]. Type 2 diabetes instruments were obtained from Xue et al.[69]. Retinal vasculature calibre instruments were obtained from Ikram et al.[4]; retinal arteriolar calibre instruments from Sim et al.[5]; and macular thickness instruments from Gao et al.[36]. Causal effects of these phenotypes on MacTel risk was assessed using logistic regression performed on unrelated individuals were the phenotypic PRS scores was regressed against disease status while correcting for sex at birth the first PC. Metabolic PRS effects on MacTel risk were corrected for multiple testing using the same strategy as described above. Potential pleiotropic effects of SNPs were tested by MR-Egger test[70].

Independent significance of metabolites as well as other phenotypes was assessed by sequential inclusion of less significant MR scores conditioning by more significant MR scores.

**MacTel PRS calculation**. Three PRS were assessed for prediction power. A top SNPs PRS was constructed by using the most significant SNP in each locus with a $P$ value less than $5 \times 10^{-8}$. SNPs that remained GW significant after conditioning for the top SNP were also included. A less stringent PRS was also constructed, using all top SNPs at loci reaching suggestive significance ($5 \times 10^{-6}$). We compared the PRSs prediction power through a logistic regression model on all unrelated individuals using the area under the ROC curve (AUC) which assess discrimination ability across different disease probability thresholds. We additionally tested for robustness of these prediction models using a leave-one-out approach applied to the entire cohort. Disease odds ratios and their 95% confidence intervals were determined for each PRS decile comparing each decile to the median. Prediction accuracy differences between European and non-Europeans were also assessed. First, the model was trained on a dataset excluding all non-European subjects as well as a randomly selected group of Europeans of the same sample size as the non-Europeans. Second, AUC of the prediction was compared for both the remaining Europeans, and non-Europeans.

**UK Biobank analysis**. Prediction of MacTel PRS was performed on 487,311 UK Biobank (UKB) subjects. UKB contains information about hospitalisations and relative primary and secondary ICD codes for admission. However, MacTel disease is unfortunately not uniquely classified by a single ICD10 code and MacTel patients are rarely hospitalised because of their condition. Because of this, we tested instead association between MacTel PRS and self-reported eye conditions using linear regression modelling. This analysis was restricted to 202,213 subjects answering the question 'Has a doctor told you that you have any of the following problems with your eyes?'. Subjects reporting 'Prefer not to answer' and 'Do not know' were excluded from the analysis.

Additionally, UKB retinal imaging grading for MacTel phenotypes was performed on optical coherence tomography by CE and TFCH using a nominal scale. Both CE and TFCH have seen several hundreds of cases of MacTel both face to face clinically and as part of their role in the Moorfields Eye Hospital Reading Centre as adjudicator of presence of disease. These subjects were randomly selected by the statistical genetics team to equally represent the top and the bottom deciles of the PRS distribution. Grading was performed on 200 subjects reporting no eye conditions. Ophthalmologists were blinded to the PRS decile for each subject. Grading was performed on the entire OCT volume scan. Each frame of the original Topcon files was converted to single.png files. The.png stacks were converted to gifs using Fiji[71]. The grading was done on the gifs, to facilitate viewing. Both eyes were graded as follows: 0 = normal, 1 = macular changes unrelated to MacTel, 2 = some macular changes resembling MacTel, 3 = eye affected by MacTel disease, NA = ungradable. Logistic regression modelling between MacTel PRS decile and maximum grading value per subject from all imaging technologies and both eyes was performed to assess correlation between MacTel PRS and retinal phenotype grading. Significance of term was assessed using likelihood-ratio tests.

**Reporting summary**. Further information on research design is available in the Nature Research Reporting Summary linked to this article.

## Data availability
Control genotyping data were sourced from dbGaP for the National Eye Institute (NEI) Age-Related Eye Disease Study (AREDS) (dbGaP accession number phs000001.v3.p1). Further control genotyping data were sourced from the QIMR Genetics of Twins study (Professor Nick Martin, available on request). MacTel patient genotyping data and a subset of the controls are available from the European Genome and Phenome archive (EGAS00001002249). Complete & European-only cohort GWAS summary statistics are available at https://doi.org/10.17605/OSF.IO/GEK7B[72].

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

## Acknowledgements

The authors would like to thank the Lowy Medical Research Institute for funding to conduct this study. We also thank Professor Nick Martin, Queensland Institute of Medical Research (QIMR), for access to the unpublished Twins Study data for controls; and Professor Stuart McGregor (QIMR) for a discussion on Mendelian Randomisation studies in cancer risk and MacTel. We thank Dr. Rinki Ratnapriya and Professor Anand Swaroop (NEI) for generously providing access to genotype data to allow targeted retinal eQTL analysis. We acknowledge the use of UK Biobank data (project application 36610) and access to the metabolomic PRS (courtesy of Dr. Claudia Langenberg, Medical Research Council, Cambridge, UK). M.B. was supported by an Australian National Health and Medical Research Council (NHMRC) Fellowship (GNT 1102971). B.R.E.A. was supported by an NHMRC Early Career Fellowship (GNT 1157776). R.B. was supported by a Melbourne International Research Scholarship and the John and Patricia Farrant Foundation. This work was also supported by the Victorian Government's Operational Infrastructure Support Program and the NHMRC Independent Research Institute Infrastructure Support Scheme (IRIISS). Finally, the authors wish to thank the MacTel patients and their families for their involvement in this study.

## Author contributions

R.B. and V.E.J.: conceptualisation, software, formal analysis, data curation, investigation, writing—original draft, writing—review and editing, and visualisation; A.P., J.E.M. and S.F.: formal analysis, data curation and investigation; N.P., C.A.E. and T.F.C.H.: resources and data curation; L.S. and R.A.: resources; M.F.: resources funding acquisition; B.R.E.A.: conceptualisation, writing—original draft, writing—review & editing, supervision project administration; M.B.: conceptualisation, writing—review & editing, supervision project administration and supervision.

## Competing interests

The authors declare no competing interests. M.B. is an Editorial Board Member for *Communications Biology*, but was not involved in the editorial review of, nor the decision to publish this article.

## Additional information

## MacTel Consortium

Rando Allikmets[7,8], Melanie Bahlo [1,2✉]

A full list of members and their affiliations also appears in Supplementary Data 16.

