## [Peer Review File · Communications Biology]

Reviewers' comments:

Reviewer #1 (Remarks to the Author):

This paper reports a comprehensive set of analyses aimed at determining the genetic risk factors for Macular Telangiectasia and elucidating the possible mechanisms of disease. The GWAS on which this study is built includes only 1067 cases with no replication cohort, however, MacTel is a rare disease and patient numbers are limiting. The GWAS is performed to a high level with all the necessary considerations for quality control. Multiple lines of evidence are given for the conclusions drawn about novel loci and collectively, this study represents a major step forward in understanding the complex etiology of this rare disease. It is unusual to see GWAS used in the study of rare disease genetics, but it is the appropriate study design for a complex disease with polygenic contributions.

The authors use a range of available gene expression, QTL and GWAS datasets to interrogate the associated genetic loci. They report on several biochemical pathways heavily implicated in this disease. While the knowledge that serine, glycine and lipid metabolism is perturbed in MacTel patients is not new, this study delves into the pathways in much more depth than previously and brings together collective information across multiple pathways, underpinned by the genetic findings. I commend the authors on their study and have only some minor questions that could provide greater clarity.

1. Table 1 presents the top ranked SNP at each genome-wide significant locus. The frequency of the effect allele (EAF) is given for each. How was the 'effect allele' defined. For some loci, this effect is protective ($OR < 1$) and for others it is increased risk ($OR > 1$). Also, for some loci it is the major allele that has the effect and for others, the minor allele.

2. The PHGDH locus is the most significantly associated. This locus was previously reported by this group in their original, smaller, GWAS for MacTel, but the two signals reported in Table 1 appear to be independent of each other, with the original SNP, rs662602 maintaining genome-wide significance after conditioning on the new top SNP, rs146953046. Could the authors comment on why the stronger signal as rs146953046 was not detected in the original GWAS?

3. Further, given that the reported effect allele of rs662602 (major allele) decreases risk, while the effect allele of rs146953046 (minor allele) increases risk, do these signals really go in the same direction when reported against the minor allele in both cases? Does the minor allele at rs146953046 occur on a haplotype with the minor allele of rs662602?

Reviewer #2 (Remarks to the Author):

This is a relevant and well-executed study. I applaud the statistical design and the manuscript reads well. The presented work significantly contributes to the field and follows up on their previous high quality genetic work. I have several questions and suggestions to further explore the data presented. I do believe, however, that reporting of additional (available) information is necessary to provide a better view on the study framework to be able to follow and understand the line of thought of the authors in the work described.

Please change the reporting of P values (capital italic and format; $P < 5.0 \times 10^{-8}$).

Please show the stratified QQ plots of the genome-wide analysis with the results stratified by frequency, imputation quality (info) score, and also consider including genomic inflation factor various frequency stratifications or imputation quality stratifications were appropriate. I was also unable to find a table with a summary of data quality control; sample-level and SNP-level data quality control, thresholds, number of removed samples and SNPs etc.

Was the current study is adequately powered to investigate associated loci. It would be good to demonstrate this by showing power for locus discovery under various genetic architectures. Including (supplemental) plots of power to discover variants at genome-wide significance for variants with ranging allele frequencies with emphasize on modestly-penetrant variants.

Despite the large sample size, it is good to consider that the absence of an independent cohort to test the observed associations across multiple cohorts is a limitation of this study and needs to be addressed in the discussion section.

Recently, GWAS studies demonstrated sexually dimorphic features of other eye conditions (Haasnoot et al. A&R 2018). Have the authors considered sex-specific association testing or used sex as an interaction term in regression analysis?

What was the diagnostic work-up for MacTel? I

In the result section, It would be helpful to visually support some of these findings and possibly conditional testing using regional plots with color-coding of the LD between with the top association(s) and the variants tested (LocusZoom).

Please provide sufficient detail to the captives of each table (i.e. abbreviations, tests etc) to be able to read and fully understand each table (currently only titles). If possible change default column names to appropriate headers I would also advice to check the captives of the main figures.

Please report statistical outcomes with uncertainty intervals were appropriate.

Adding annotation of relevant peaks and appropriate size of the font will make Figure 1 more informative. Also, I personally believe some of the figures can be improved to make them more appealing.

Could the authors speculate on the genetic eQTL effects outside the retina that contribute to disease?

Reviewer #3 (Remarks to the Author):

This manuscript details novel genomic findings on MacTel, a rare retinal disorder. This study not only corroborates prior findings but identifies several new loci and potential disease mechanisms on the largest MacTel study to date (n = 1,067 MacTel patients; n=3,779 controls). The statistical and bioinformatic analysis are elegant, well executed, and include validation of findings in retinal tissues from the eye to help support disease mechanisms governing the pathophysiology of this disease.

Some very minor points to address to make the work presented here clearer to interpretation:

1. It appears that this is a separate MacTel study (different sets of cases and controls) than the prior GWAS study published in 2017? Is this correct? There is a line in the discussion which states that this is not an independent study. Please clearly state in results and discussion how this GWAS relates to the prior GWAS (in terms of subject overlap or not), in addition to corroboration of findings.

2. In the results it would be helpful to write a short sentence on what constituted proximal eg., in number of bases. This will help the reader to better appreciate how the 52 genes were identified

proximal to the associated loci.

3. Also in the results section may want to remind the reader what "at least two sources" are for evidence are for gene prioritization.

Reviewer #4 (Remarks to the Author):

Bonelli R et al, conducted a more expansive GWAS study using an expanded number of cases and controls from their initial study in 2017. All three previously discovered GWAS loci reached genome-wide significance, while novel GWAS loci was also discovered to significantly associate with Mactel risk . These GWAS data were then used in meta-analysis using other databases to infer pathways that could drive Mactel pathogenesis. Unsurprisingly and in agreement with prior and upcoming metabolomic GWAS studies on Mactel, the de novo serine glycine synthesis pathway emerged as a key candidate for Mactel risk. A polygenic risk model based on GWAS SNP was generated and applied onto the UK biobank data, where it was positively associated with eye disorders.

Overall, the treatment of GWAS data and its associated conclusions appears to be rigorous and sound. However, the subsequent conclusions based off data mining studies appears to be merely confirming what has been found in the literature: that serine glycine synthesis pathway is a major risk factor in Mactel (as referenced by the authors themselves). Thus, despite the strength of the GWAS data, it is unclear how the major conclusions of this paper add anything to what is already known about Mactel pathogenesis.

All prior data from other studies consist of inferences based upon patient GWAS data and public databases. There is still no direct mechanistic evidence illuminating the role of serine/glycine biosynthesis in Mactel pathogenesis. Thus, this reviewer suggests that the authors utilize actual Mactel disease tissue or derived patient material to generate actual mechanistic insight into Mactel pathogenesis. This will add substantial value to this manuscript that is sufficient for acceptance by this journal.

Specific comments:

1. rs146953046 is independent from the previously discovered rs477992 within the PHGDH locus. While rs477992 is protective, rs146953046 is a strong risk allele (OR= 5.47) even though it is a predicted benign allele in Clinvar, and is not expected to change the 5' splice site strength of the upstream exon. The authors should directly investigate expression differences between the two SNPs, and also whether there are differences in PHGDH expression in Mactel retina vs control patient tissue. If possible, expression of PHGDH in Mactel retina or immortalized lymphocytes bearing rs146953046 and rs477992 should be compared. This is an extremely key point in this paper because PHGDH is the key rate limiting enzyme for serine/glycine de novo biosynthesis, yet it is still unclear from eQTL or TWAS data (see Pt. 2) the direction of PHGDH expression alteration that could be associated with Mactel.

2. "PHGDH and MXRA7, and down-regulation of 188 nuclear receptor binding factor 2 (NRBF2), was inferred to take place in MacTel patient retinae" (Line 187-188). How does increased PHGDH expression lead to decreased systemic serine serum concentration in Mactel patients? Importantly, how does this dovetail with an earlier claim "Interestingly strong suppression of 176 PHGDH (1p12) transcription was inferred in the tibial nerve, brain and vasculature, but not in the retina." What exactly are the authors claiming here with respect to PHGDH expression in Mactel tissue?

3. On a similar vein, PSPH expression in Mactel vs. Control tissues should be investigated, since PSPH is the terminal enzyme in serine synthesis.

4. rs2160387 and rs17279437 implicated serine/glycine transporters with Mactel risk. These are key genes that need to be investigated, since they are involved in secretion/uptake of serine and/or glycine, thus contributing to alterations of serum concentrations of serine and glycine that seem to be

key to Mactel pathogenesis. Are these expressed only in retina, or systemically, and are their abundances altered in Mactel? Some data here will greatly provide insight on systemic and/or ocular contribution of altered serine glycine metabolism in Mactel.

Bonelli et al Response to Reviewer Comments

We thank the reviewers for their close reading of the manuscript and helpful comments and suggestions, which we address below. In the current submission we include an updated version of the manuscript with changes tracked for the reviewers' information.

We note that the tagging SNP chr1:120265444 (GRCh37) was incorrectly quoted as 'rs662602' in our original submission. It has been updated to rs532303. For consistency we have updated the relevant rsID in both the reviewer comments, our responses, and the revised manuscript.

Reviewer #1

This paper reports a comprehensive set of analyses aimed at determining the genetic risk factors for Macular Telangiectasia and elucidating the possible mechanisms of disease. The GWAS on which this study is built includes only 1067 cases with no replication cohort, however, MacTel is a rare disease and patient numbers are limiting. The GWAS is performed to a high level with all the necessary considerations for quality control. Multiple lines of evidence are given for the conclusions drawn about novel loci and collectively, this study represents a major step forward in understanding the complex etiology of this rare disease. It is unusual to see GWAS used in the study of rare disease genetics, but it is the appropriate study design for a complex disease with polygenic contributions.

The authors use a range of available gene expression, QTL and GWAS datasets to interrogate the associated genetic loci. They report on several biochemical pathways heavily implicated in this disease. While the knowledge that serine, glycine and lipid metabolism is perturbed in MacTel patients is not new, this study delves into the pathways in much more depth than previously and brings together collective information across multiple pathways, underpinned by the genetic findings.

I commend the authors on their study and have only some minor questions that could provide greater clarity.

1. Table 1 presents the top ranked SNP at each genome-wide significant locus. The frequency of the effect allele (EAF) is given for each. How was the 'effect allele' defined. For some loci, this effect is protective ($OR < 1$) and for others it is increased risk ($OR > 1$). Also, for some loci it is the major allele that has the effect and for others, the minor allele.

Response 1. We thank the reviewer for their recognition of the advances in MacTel genetics that our paper reports, and we are pleased that they found our manuscript sound and informative. Throughout this manuscript we opted to use effect/non-effect allele terminology, to remove any ambiguity as to which allele the reported odds ratios relates to. This approach is recommended in doi.org/10.1093/ije/dyaa149. Throughout, the effect allele is always the non-reference allele. We have added this clarification to the legend of relevant tables.

2. The PHGDH locus is the most significantly associated. This locus was previously reported by this group in their original, smaller, GWAS for MacTel, but the two signals reported in Table 1 appear to be independent of each other, with the original SNP, rs532303 maintaining genome-wide significance after conditioning on the new top SNP, rs146953046. Could the authors comment on why the stronger signal as rs146953046 was not detected in the original GWAS?

Response 2. The SNP rs146953046 (MAF = 1.6% in controls) was not included in the original GWAS because SNPs with MAF less than 5% were excluded from the analyses. This decision to exclude such SNPs was based on a) the modest sample size of the first GWAS study, which did not provide sufficient power to detect associations for SNPs with MAF < 5%; and b) the need to limit false positive results. The present manuscript benefits from both a larger sample size and improvements in imputation panels, which can better infer haplotypes. Thus imputed SNPs are more robust. In the revised manuscript we now include both the results of our power calculations (detailed below), and discussion of the benefits of improved imputation methods.

3. Further, given that the reported effect allele of r662602 (major allele) decreases risk, while the effect allele of rs146953046 (minor allele) increases risk, do these signals really go in the same direction when reported against the minor allele in both cases? Does the minor allele at rs146953046 occur on a haplotype with the minor allele of rs532303?

Response 3. We confirm that it is indeed the case that both minor alleles are the risk alleles (ie G allele for rs146953046; A allele for rs532303). We note that these two SNPs are not in strong linkage disequilibrium in the 1000 Genomes European population (SNPs $R^2 = 0.001$; $D' = 0.2036$). Within our data, the haplotypes in cases and controls are distributed as follows:

Controls:

		rs146953046		Total (AF)
		T	G	
rs532303	G	5141	64	5205 (0.69)
	A	2334	59	2393 (0.31)
Total (AF)		7475 (0.984)	123 (0.016)	

Cases:

		rs146953046		Total (AF)
		T	G	
rs532303	G	1156	71	1227 (0.58)
	A	846	60	906 (0.42)
Total (AF)		2002 (0.94)	131 (0.06)	

In both cases and controls, these SNPs are not in strong linkage disequilibrium, particularly in cases (controls $r^2=0.048$; $D'=0.254$; cases: $r^2=0.025$; $D'=0.084$).

Furthermore, the observed associations for rs532303 were very similar, in the main analysis, and in the conditional analysis, where rs146953046 was included as a covariate ($p_{\text{unconditional}}=2.38 \times 10^{-20}$; $p_{\text{conditional}}=9.29 \times 10^{-18}$). We have now included this information in Table S4.

Reviewer #2

This is a relevant and well-executed study. I applaud the statistical design and the manuscript reads well. The presented work significantly contributes to the field and follows up on their previous high-quality genetic work. I have several questions and suggestions to further explore the data presented. I do believe, however, that reporting of additional (available) information is necessary to provide a better view on the study framework to be able to follow and understand the line of thought of the authors in the work described.

1. Please change the reporting of P values (capital italic and format; $P < 5.0 \times 10^{-8}$).

Response 1. We thank the reviewer for their compliments on our work. We have now changed the P value reporting throughout the manuscript as requested.

2. Please show the stratified QQ plots of the genome-wide analysis with the results stratified by frequency, imputation quality (info) score, and also consider including genomic inflation factor various frequency stratifications or imputation quality stratifications were appropriate. I was also unable to find a table with a summary of data quality control; sample-level and SNP-level data quality control, thresholds, number of removed samples and SNPs etc.

Response 2. New QQ-plots have been included in Figure S4, stratified by MAF (<5% and ≥5%), and with different INFO score filters (≥0.9 and ≥0.95). Stratified Lambda values have been added to the captions of this figure. Full details of sample and SNP exclusions have now been included in supplementary Table S1.

3. Was the current study is adequately powered to investigate associated loci. It would be good to demonstrate this by showing power for locus discovery under various genetic architectures. Including (supplemental) plots of power to discover variants at genome-wide significance for variants with ranging allele frequencies with emphasize on modestly-penetrant variants.

Response 3. Assuming a population prevalence of MacTel of 0.45%, and an additive genetic model, we had 80% power to detect genome-wide significant associations with odds ratio (OR) of 1.5 for SNPs with population effect allele (EA) frequencies (EAF) greater than 20%; OR=2.0 for SNPs with EAF greater than 5%; and OR=4.0 for SNPs with EAF greater than 1%. Effects of this magnitude were plausible, given the associations we identified in our previous work (Scerri et al. 2017). We have now included a plot of power vs EAF, for varying ORs in the supplement (Figure S2).

4. *Despite the large sample size, it is good to consider that the absence of an independent cohort to test the observed associations across multiple cohorts is a limitation of this study and needs to be addressed in the discussion section.*

Response 4. We agree the absence of a replication cohort is a limitation of this study. Given that MacTel is a rare disease (prevalence of ~4/1,000), an independent replication cohort is not presently available. We now include discussion of the implications of the lack of a replication cohort.

5. *Recently, GWAS studies demonstrated sexually dimorphic features of other eye conditions (Haasnoot et al. A&R 2018). Have the authors considered sex-specific association testing or used sex as an interaction term in regression analysis?*

Response 5. We thank the reviewer for this interesting suggestion, and agree it is a particularly pertinent question for MacTel, given the sex-interaction previously observed at the 2q34 (*CPS1*) locus (Scerri et al. 2017). For the 11 top SNPs, we undertook sex-specific association analyses, and formally tested for an interaction between the male and female effects.

A significant sex-interaction was found for rs1047891 at 2q34 ($P_{\text{interaction}}=1.06 \times 10^{-4}$), with a greater effect seen in females compared to males. Nominally significant interactions were observed for the two SNPs in the 1p12 locus, again with a larger effect in females. We have now included the results of these analyses in the main manuscript sub-section title 'Sex Interaction Analysis,' and relevant results are given in Table S5 and Figure S6.

6. *What was the diagnostic work-up for MacTel?*

Response 6. We have now included a paragraph in Supplementary Methods to describe the MacTel diagnostic procedures.

7. *In the result section, It would be helpful to visually support some of these findings and possibly conditional testing using regional plots with color-coding of the LD between with the top association(s) and the variants tested (LocusZoom).*

Response 7. We thank the reviewer for this suggestion, and we now provide three LocusZoom region plots for the 1p12 signals, in Figure S5 These added plots are as follows:

A) 1p12 signal 1: Top SNP rs146953046 (chr1:120278072) - P-values conditional on rs146953046 (chr1:120265444).

B) 1p12 signal 2: Top SNP rs146953046 (chr1:120265444) - P-values conditional on rs146953046 (chr1:120278072).

C) 1p12 region with two independent signals: rs146953046 (chr1:120278072) and rs146953046 (chr1:120278072). Non-conditional P-values shown. SNPs in LD with rs146953046

(chr1:120278072) coloured in blue; SNPs in LD with rs146953046 (chr1:120265444) coloured in red.

8. Please provide sufficient detail to the captions of each table (i.e. abbreviations, tests etc) to be able to read and fully understand each table (currently only titles). If possible change default column names to appropriate headers I would also advice to check the captions of the main figures.

Response 8. We have now included additional definitions in the supplementary table legends, and figure captions.

9. Please report statistical outcomes with uncertainty intervals were appropriate.

Response 9. Where possible and informative confidence intervals were included (e.g. Table 1, Table 2 and AUC estimation). For other analysis these intervals were either not-available (eQTL, TWAS, MAGMA, Colocalization) or/and not particularly informative as the direction of effect, rather than effect size, was interpreted (MR, UKB analysis).

10. Adding annotation of relevant peaks and appropriate size of the font will make Figure 1 more informative. Also, I personally believe some of the figures can be improved to make them more appealing.

Response 10. Due to space and font size limitations in Figure 1, in the legend we now refer readers to Tables 1 and 2 for full details of the 17 MacTel loci of interest.

11. Could the authors speculate on the genetic eQTL effects outside the retina that contribute to disease?

Response 11. We agree that extra-retinal eQTLs are an interesting facet of MacTel genetics and strongly suggest a systemic underpinning for this disease. To better delineate the potential contributions of different body tissues to MacTel, in response to this comment and Reviewer 4 comment 4 (please see below), we have curated the tissue-specific expression of MacTel genes in several tissues (Figure S9 and S10) and revised the discussion accordingly.

Reviewer #3

This manuscript details novel genomic findings on MacTel, a rare retinal disorder. This study not only corroborates prior findings but identifies several new loci and potential disease mechanisms on the largest MacTel study to date (n = 1,067 MacTel patients; n=3,779 controls). The statistical and bioinformatic analysis are elegant, well executed, and include validation of findings in retinal tissues from the eye to help support disease mechanisms governing the pathophysiology of this disease.

Some very minor points to address to make the work presented here clearer to interpretation:

1. It appears that this is a separate MacTel study (different sets of cases and controls) than the prior GWAS study published in 2017? Is this correct? There is a line in the discussion which states that this is not an independent study. Please clearly state in results and discussion how this GWAS relates to the prior GWAS (in terms of subject overlap or not), in addition to corroboration of findings.

Response 1. We thank the reviewer for their compliments on our work.

We have now amended the results and discussion section to clarify the subject and consequential results overlap between the original GWAS (Scerri et al, Nat Genet 2017) and the new GWAS, described in this work.

2. In the results it would be helpful to write a short sentence on what constituted proximal eg., in number of bases. This will help the reader to better appreciate how the 52 genes were identified proximal to the associated loci.

Response 2. We have now clarified this point. Forty-eight genes were prioritised because SNPs in LD with the main SNP of interest at each locus were within gene regions. Only four proximal genes were defined for two intergenic genome-wide significant loci (two genes respectively) for which no SNPs in LD were found to overlap any protein-coding gene. These four genes were hence defined as the closest upstream and downstream genes to these two genome wide-significant SNPs. We have now included a sentence in the 'Gene Prioritization' results section to clarify this.

3. Also in the results section may want to remind the reader what "at least two sources" are for evidence are for gene prioritization.

Response 3. We have now clarified the evidence sources in the results sub-section entitled 'Gene expression patterns across body tissues and retinal cell-types'.

Reviewer #4

Bonelli R et al, conducted a more expansive GWAS study using an expanded number of cases and controls from their initial study in 2017. All three previously discovered GWAS loci reached genome-wide significance, while novel GWAS loci was also discovered to significantly associate with Mactel risk. These GWAS data were then used in meta-analysis using other databases to infer pathways that could drive Mactel pathogenesis. Unsurprisingly and in agreement with prior and upcoming metabolomic GWAS studies on Mactel, the de novo serine glycine synthesis pathway emerged as a key candidate for Mactel risk. A polygenic risk model based on GWAS SNP was generated and applied onto the UK biobank data, where it was positively associated with eye disorders.

Overall, the treatment of GWAS data and its associated conclusions appears to be rigorous and sound. However, the subsequent conclusions based off data mining studies appears to be merely confirming what has been found in the literature: that serine glycine synthesis pathway is a major risk factor in Mactel (as referenced by the authors themselves). Thus, despite the strength of the GWAS data, it is unclear how the major conclusions of this paper add anything to what is already known about Mactel pathogenesis.

All prior data from other studies consist of inferences based upon patient GWAS data and public databases. There is still no direct mechanistic evidence illuminating the role of serine/glycine biosynthesis in Mactel pathogenesis. Thus, this reviewer suggests that the authors utilize actual Mactel disease tissue or derived patient material to generate actual mechanistic insight into Mactel pathogenesis. This will add substantial value to this manuscript that is sufficient for acceptance by this journal.

Response

We thank the reviewer for their careful reading of the manuscript and suggestions for improvement. However we respectfully disagree with the claims that this study is merely confirming existing knowledge about the genetics of MacTel. We also contest the idea that MacTel is entirely a glycine-serine metabolic disorder. In assembling the largest cohort of MacTel cases analysed to date, we report eight new genetic risk factors involved in biological processes as diverse as vascular matrix integrity, lipid metabolism, and amino acid transport, in addition to confirming the role of glycine-serine dysregulation in this disease. We further report the most powerful polygenic risk predictor for MacTel to date and prove its effectiveness for population disease inference in the UK Biobank. In light of the Reviewer's comments, we now also provide a putative mechanism for the effect of the newly discovered rare SNP on *PHGDH* expression in the retina, and throughout the body (below).

The major value of our paper lies in the wealth of new hypotheses regarding the systemic and tissue-specific aspects of MacTel, supported with results that leverage large independently-generated functional genomics experiments (GTEx, eBIOs, FUMA) and clinical datasets (UK Biobank). Further, the molecular pathways and genetic etiological insights we describe were derived via a rigorous, unbiased and data-driven approach (unlike many targeted functional studies). As our work makes clear, MacTel is not solely a metabolic disorder, but instead implicates several metabolic, developmental, structural and tissue-specific biological changes that converge on retinal dysfunction. As per our initial genome-wide investigation of MacTel (Scerri et al *Nat Genet* 2017) which drove subsequent studies validating our molecular hypotheses (Gantner et al *NEJM* 2019), we expect that targeted functional studies of the newly reported genetic signals will be illuminating, but they are the domain of future papers.

Specific comments:

1. rs146953046 is independent from the previously discovered rs477992 within the PHGDH locus. While rs477992 is protective, rs146953046 is a strong risk allele (OR = 5.47) even though it is a predicted benign allele in Clinvar, and is not expected to change the 5' splice site strength of the upstream exon. The authors should directly investigate expression differences between the two SNPs, and also whether there are differences in PHGDH expression in Mactel retina vs control patient tissue. If possible, expression of PHGDH in Mactel retina or immortalized lymphocytes bearing rs146953046 and rs477992 should be compared. This is an extremely key point in this paper because PHGDH is the key rate limiting enzyme for serine/glycine de novo biosynthesis, yet it is still unclear from eQTL or TWAS data (see Pt. 2) the direction of PHGDH expression alteration that could be associated with Mactel.

Response 1.

Regarding the disease risk conferred by the rare and common SNPs localized to *PHGDH*, we refer the Reviewer to our response to Reviewer 1 comment 3, where we performed linkage disequilibrium analysis and clarified the effect alleles (please see above). Unfortunately MacTel patient tissue is exceedingly rare, and thus experiments using patient tissue are well beyond the scope of this work. Further, we contend that experiments using patient-derived lymphoblasts, even if available, would be of limited use for understanding the complex contributions of *PHGDH* to MacTel, as this cell type is not central to the disease as currently understood. Instead, *PHGDH* expression in retinal tissues and the liver (a major site of serine and sphingolipid metabolism), are of primary concern for our work.

Therefore, to better understand the role of the rare deleterious SNP rs146953046, we collaborated with the US National Eye Institute, making use of the data in Ratnapriya et al, *Nat Genet* 2019, to impute this SNP, and examine it for the first time in 406 subjects for whom retinal RNAseq was available. This identified 9 heterozygotes (2 healthy controls and 7 AMD patients). Correcting for age, sex and AMD disease status, we identified a significant suppressive eQTL effect of rs146953046 on

PHGDH expression in the retina. Indeed, this agrees with findings in the GTEx cohort for rs146953046, in which there are 27 tissues with significant negative eQTL effects, but none with positive effects. These findings are now included in Figure S8 panels A and B. Regarding the potential splicing effects of this SNP, although there are four tissues in GTEx for which rs146953046 is both an eQTL and a spliceQTL, we performed differential exon usage analysis using a similar statistical framework as for eQTL analysis and found no evidence for a splicing effect in the retina (Figure S8C), but we note that the current low read numbers in retinal eQTL data do not permit a well powered analysis for splicing effects with currently available data.

Together these findings provide strong evidence that rs146953046 is likely disease-causing, suppressing serine biosynthesis in the retina and throughout the body. The precise mechanism remains to be determined but is likely mediated through reduced expression. This finding agrees with lower circulating serine reported in MacTel patients, and subsequent accumulation of retinotoxic deoxysphingolipids (Gantner et al 2019, *NEJM*).

2. *“Increased expression of PHGDH and MXRA7, and down-regulation of nuclear receptor binding factor 2 (NRBF2), was inferred to take place in MacTel patient retinae” (Line 187-188). How does increased PHGDH expression lead to decreased systemic serine serum concentration in Mactel patients? Importantly, how does this dovetail with an earlier claim “Interestingly strong suppression of PHGDH (1p12) transcription was inferred in the tibial nerve, brain and vasculature, but not in the retina.” What exactly are the authors claiming here with respect to PHDGH expression in Mactel tissue?*

Response 2.

These statements pertain to TWAS results and eQTL results respectively. We have updated this text in light of the new rs146953046 eQTL effects on *PHGDH* identified in the retina. However it is worth noting that the new analysis, which identified the suppressive eQTL result for SNP rs146953046, is not directly comparable with the pre-existing genome-wide retinal eQTL results for statistical reasons. We have now revised the result to “strong suppression of *PHGDH* (1p12) transcription was inferred in the tibial nerve, brain and vasculature.”

Conversely, the TWAS analysis suggested increased expression of *PHGDH* in the retina of MacTel patients as the estimated net effect of all eQTLs impinging on that gene (spanning 95,815 base pairs). The rare SNP rs146953046 eQTL was not included in the summary statistics published by Ratnapriya *et al* used in our TWAS, which is a limitation of the current study and noted in the discussion. It was not possible to revise the TWAS analysis to include rs146953046, due to the differences in analyses, as noted above. We also caution interpretation of the TWAS result in the neural retina as the RPE may be a more relevant tissue for serine and sphingolipid metabolism, which are then supplied to the retina via transporters including SLC family genes. Unfortunately no eQTL datasets are available for

the adult RPE to examine these effects in more detail. We emphasize the need for further experimental investigation into the effects of *PHGDH* variants in a tissue-specific manner.

3. *On a similar vein, PSPH expression in Mactel vs. Control tissues should be investigated, since PSPH is the terminal enzyme in serine synthesis.*

Response 3.

We have included investigation of *PSPH* in multiple control tissues relevant for this disease as part of the response to comment 4 (below).

4. *rs2160387 and rs17279437 implicated serine/glycine transporters with Mactel risk. These are key genes that need to be investigated, since they are involved in secretion/uptake of serine and/or glycine, thus contributing to alterations of serum concentrations of serine and glycine that seem to be key to Mactel pathogenesis. Are these expressed only in retina, or systematically, and are their abundances altered in Mactel? Some data here will greatly provide insight on systemic and/or ocular contribution of altered serine glycine metabolism in Mactel.*

Response 4.

We agree that the potential for tissue-specific expression of SLC transporters implicated in transport of metabolites central for MacTel is of great interest. Additionally, this class of proteins are highly druggable [<https://doi.org/10.1038/nrd4626>]. To address this important point we ranked the expression in six tissues of 49 genes prioritized through post-GWAS analyses (≥ 2 lines of evidence; including *SLC1A4*, *SLC6A20* and *SLC4A7*), and added six SLCs supported by at least one line of evidence (Table S11; Figure S9). This allows a fair comparison of gene expression between different tissues from different studies, and will provide readers with a useful resource for comparing genetic contributions to MacTel from different tissues. We also updated Figure S10 and Tables S12 and S13 to include comparison of SLCs between the neural retina and retinal pigment epithelium, and between single retinal cell types. For example, whereas *SLC4A7* is the most highly expressed SLC in the neural retina (Figure S9), lowly expressed genes in that tissue such as SLCs *6A20* and *16A8* are preferentially expressed in the retinal pigment epithelium (Figure S10). We have revised the discussion in light of this analysis and thank the reviewer for the suggestion.

REVIEWERS' COMMENTS:

Reviewer #1 (Remarks to the Author):

All my questions have been answered

Reviewer #2 (Remarks to the Author):

The authors have carefully addressed all my concerns and requests.

Reviewer #3 (Remarks to the Author):

The authors have done an excellent job of addressing the reviewer's concerns.

Reviewer #4 (Remarks to the Author):

Reviewer #4: Bonelli R et al, conducted a more expansive GWAS study using an expanded number of cases and controls from their initial study in 2017. All three previously discovered GWAS loci reached genomewide significance, while novel GWAS loci was also discovered to significantly associate with Mactel risk. These GWAS data were then used in meta-analysis using other databases to infer pathways that could drive Mactel pathogenesis. Unsurprisingly and in agreement with prior and upcoming metabolomic GWAS studies on Mactel, the de novo serine glycine synthesis pathway emerged as a key candidate for Mactel risk. A polygenic risk model based on GWAS SNP was generated and applied onto the UK biobank data, where it was positively associated with eye disorders. Overall, the treatment of GWAS data and its associated conclusions appears to be rigorous and sound. However, the subsequent conclusions based off data mining studies appears to be merely confirming what has been found in the literature: that serine glycine synthesis pathway is a major risk factor in Mactel (as referenced by the authors themselves). Thus, despite the strength of the GWAS data, it is unclear how the major conclusions of this paper add anything to what is already known about Mactel pathogenesis. All prior data from other studies consist of inferences based upon patient GWAS data and public databases. There is still no direct mechanistic evidence illuminating the role of serine/glycine biosynthesis in Mactel pathogenesis. Thus, this reviewer suggests that the authors utilize actual Mactel disease tissue or derived patient material to generate actual mechanistic insight into Mactel pathogenesis. This will add substantial value to this manuscript that is sufficient for acceptance by this journal.

Authors Response: We thank the reviewer for their careful reading of the manuscript and suggestions for improvement. However we respectfully disagree with the claims that this study is merely confirming existing knowledge about the genetics of MacTel. We also contest the idea that MacTel is entirely a glycineserine metabolic disorder. In assembling the largest cohort of MacTel cases analysed to date, we report eight new genetic risk factors involved in biological processes as diverse as vascular matrix integrity, lipid metabolism, and amino acid transport, in addition to confirming the role of glycine-serine dysregulation in this disease. We further report the most powerful polygenic risk predictor for MacTel to date and prove its effectiveness for population disease inference in the UK Biobank. In light of the Reviewer's comments, we now also provide a putative mechanism for the effect of the newly discovered rare SNP on PHGDH expression in the retina, and throughout the body (below). The major value of our paper lies in the wealth of new hypotheses regarding the systemic and tissuespecific aspects of MacTel, supported with results that leverage large independently-

generated functional genomics experiments (GTEx, eBIOS, FUMA) and clinical datasets (UK Biobank). Further, the molecular pathways and genetic etiological insights we describe were derived via a rigorous, unbiased and data-driven approach (unlike many targeted functional studies). As our work makes clear, MacTel is not solely a metabolic disorder, but instead implicates several metabolic, developmental, structural and tissue-specific biological changes that converge on retinal dysfunction. As per our initial genome-wide investigation of MacTel (Scerri et al Nat Genet 2017) which drove subsequent studies validating our molecular hypotheses (Gantner et al NEJM 2019), we expect that targeted functional studies of the newly reported genetic signals will be illuminating, but they are the domain of future papers.

Reviewer's Response: I fully acknowledge the importance of the new hypotheses generated from functional genomics datasets, and I completely agree that they are rigorous and unbiased. I have no doubt that they will serve as excellent starting points for a full-fledged mechanistic investigation of MacTel pathogenesis. However, my intention in asking for functional data based off Mactel tissue is such that some of the key findings could at least be validated in the relevant tissue in order to complement and strengthen the data which has been generated from bioinformatic datasets. Specifically, this pertains to rs146953046 and its effects on PHGDH expression. Indeed, the authors have now generated new data that suggests that rs146953046 is suppressive to PHGDH expression. Though I believe these data have adequately answered my comments, I am still of the opinion that even though it is hard to study rs146953046 in Mactel tissue, the data can still be validated in RPE and liver cell lines, which are commercially available. Doing so will definitively confirm the effect of rs146953046, validate the bioinformatic findings and elevate the value of the manuscript.

Specific comments:

1. rs146953046 is independent from the previously discovered rs477992 within the PHGDH locus. While rs477992 is protective, rs146953046 is a strong risk allele (OR = 5.47) even though it is a predicted benign allele in Clinvar, and is not expected to change the 5' splice site strength of the upstream exon. The authors should directly investigate expression differences between the two SNPs, and also whether there are differences in PHGDH expression in Mactel retina vs control patient tissue. If possible, expression of PHGDH in Mactel retina or immortalized lymphocytes bearing rs146953046 and rs477992 should be compared. This is an extremely key point in this paper because PHGDH is the key rate limiting enzyme for serine/glycine de novo biosynthesis, yet it is still unclear from eQTL or TWAS data (see Pt. 2) the direction of PHGDH expression alteration that could be associated with Mactel.

Author's Response 1. Regarding the disease risk conferred by the rare and common SNPs localized to PHGDH, we refer the Reviewer to our response to Reviewer 1 comment 3, where we performed linkage disequilibrium analysis and clarified the effect alleles (please see above). Unfortunately MacTel patient tissue is exceedingly rare, and thus experiments using patient tissue are well beyond the scope of this work. Further, we contend that experiments using patient-derived lymphoblasts, even if available, would be of limited use for understanding the complex contributions of PHGDH to MacTel, as this cell type is not central to the disease as currently understood. Instead, PHGDH expression in retinal tissues and the liver (a major site of serine and sphingolipid metabolism), are of primary concern for our work. Therefore, to better understand the role of the rare deleterious SNP rs146953046, we collaborated with the US National Eye Institute, making use of the data in Ratnapriya et al, Nat Genet 2019, to impute this SNP, and examine it for the first time in 406 subjects for whom retinal RNAseq was available. This identified 9 heterozygotes (2 healthy controls and 7 AMD patients). Correcting for age, sex and AMD disease status, we identified a significant suppressive eQTL effect of rs146953046 on PHGDH expression in the retina. Indeed, this agrees with findings in the

GTEx cohort for rs146953046, in which there are 27 tissues with significant negative eQTL effects, but none with positive effects. These findings are now included in Figure S8 panels A and B. Regarding the potential splicing effects of this SNP, although there are four tissues in GTEx for which rs146953046 is both an eQTL and a spliceQTL, we performed differential exon usage analysis using a similar statistical framework as for eQTL analysis and found no evidence for a splicing effect in the retina (Figure S8C), but we note that the current low read numbers in retinal eQTL data do not permit a well powered analysis for splicing effects with currently available data. Together these findings provide strong evidence that rs146953046 is likely disease-causing, suppressing serine biosynthesis in the retina and throughout the body. The precise mechanism remains to be determined but is likely mediated through reduced expression. This finding agrees with lower circulating serine reported in MacTel patients, and subsequent accumulation of retinotoxic deoxysphingolipids (Gantner et al 2019, NEJM).

Reviewer's Response: Given the scarcity of Mactel tissue, the new data included in the Revision has adequately answered my comments.

2. "Increased expression of PHGDH and MXRA7, and down-regulation of nuclear receptor binding factor 2 (NRBF2), was inferred to take place in MacTel patient retinae" (Line 187-188). How does increased PHGDH expression lead to decreased systemic serine serum concentration in Mactel patients? Importantly, how does this dovetail with an earlier claim "Interestingly strong suppression of PHGDH (1p12) transcription was inferred in the tibial nerve, brain and vasculature, but not in the retina." What exactly are the authors claiming here with respect to PHGDH expression in Mactel tissue?

Response 2. These statements pertain to TWAS results and eQTL results respectively. We have updated this text in light of the new rs146953046 eQTL effects on PHGDH identified in the retina. However it is worth noting that the new analysis, which identified the suppressive eQTL result for SNP rs146953046, is not directly comparable with the pre-existing genome-wide retinal eQTL results for statistical reasons. We have now revised the result to "strong suppression of PHGDH (1p12) transcription was inferred in the tibial nerve, brain and vasculature." Conversely, the TWAS analysis suggested increased expression of PHGDH in the retina of MacTel patients as the estimated net effect of all eQTLs impinging on that gene (spanning 95,815 base pairs). The rare SNP rs146953046 eQTL was not included in the summary statistics published by Ratnapriya et al used in our TWAS, which is a limitation of the current study and noted in the discussion. It was not possible to revise the TWAS analysis to include rs146953046, due to the differences in analyses, as noted above. We also caution interpretation of the TWAS result in the neural retina as the RPE may be a more relevant tissue for serine and sphingolipid metabolism, which are then supplied to the retina via transporters including SLC family genes. Unfortunately no eQTL datasets are available for the adult RPE to examine these effects in more detail. We emphasize the need for further experimental investigation into the effects of PHGDH variants in a tissue-specific manner.

Reviewer's Response: The Authors' new data in Point 1 strengthens the evidence that PHGDH suppression and by implication, lower circulating serine could likely be contributing towards Mactel risk and onset. I completely agree with the Authors that the TWAS and eQTL data is not directly compatible. Here, my concern is that TWAS data could be distracting since it appears to be contradicting the major point of this manuscript. Hence I would like to suggest that the TWAS data be removed from the manuscript, if possible.

3. On a similar vein, PSPH expression in Mactel vs. Control tissues should be investigated, since PSPH is the terminal enzyme in serine synthesis.

Author's Response 3. We have included investigation of PSPH in multiple control tissues relevant for this disease as part of the response to comment 4 (below).

4. rs2160387 and rs17279437 implicated serine/glycine transporters with Mactel risk. These are key genes that need to be investigated, since they are involved in secretion/uptake of serine and/or glycine, thus contributing to alterations of serum concentrations of serine and glycine that seem to be key to Mactel pathogenesis. Are these expressed only in retina, or systematically, and are their abundances altered in Mactel? Some data here will greatly provide insight on systemic and/or ocular contribution of altered serine glycine metabolism in Mactel.

Response 4. We agree that the potential for tissue-specific expression of SLC transporters implicated in transport of metabolites central for MacTel is of great interest. Additionally, this class of proteins are highly druggable [<https://doi.org/10.1038/nrd4626>]. To address this important point we ranked the expression in six tissues of 49 genes prioritized through post-GWAS analyses (≥ 2 lines of evidence; including SLC1A4, SLC6A20 and SLC4A7), and added six SLCs supported by at least one line of evidence (Table S11; Figure S9). This allows a fair comparison of gene expression between different tissues from different studies, and will provide readers with a useful resource for comparing genetic contributions to MacTel from different tissues. We also updated Figure S10 and Tables S12 and S13 to include comparison of SLCs between the neural retina and retinal pigment epithelium, and between single retinal cell types. For example, whereas SLC4A7 is the most highly expressed SLC in the neural retina (Figure S9), lowly expressed genes in that tissue such as SLCs 6A20 and 16A8 are preferentially expressed in the retinal pigment epithelium (Figure S10). We have revised the discussion in light of this analysis and thank the reviewer for the suggestion.

Reviewer's Response to 3 and 4: The new data has more than answered my comments. Thank you!